# C53 Interacting with UFM1-Protein Ligase 1 Regulates Microtubule Nucleation in Response to ER Stress

**DOI:** 10.3390/cells11030555

**Published:** 2022-02-05

**Authors:** Anastasiya Klebanovych, Stanislav Vinopal, Eduarda Dráberová, Vladimíra Sládková, Tetyana Sulimenko, Vadym Sulimenko, Věra Vosecká, Libor Macůrek, Agustin Legido, Pavel Dráber

**Affiliations:** 1Laboratory of Biology of Cytoskeleton, Institute of Molecular Genetics of the Czech Academy of Sciences, CZ 142 20 Prague, Czech Republic; AKlebanovych@danforthcenter.org (A.K.); stanislav.vinopal@ujep.cz (S.V.); Eduarda.Draberova@img.cas.cz (E.D.); vladimira.sladkova@img.cas.cz (V.S.); tetyana.sulimenko@img.cas.cz (T.S.); vadym.sulimenko@img.cas.cz (V.S.); vera.vosecka@img.cas.cz (V.V.); libor.macurek@img.cas.cz (L.M.); 2Section of Neurology, St. Christopher’s Hospital for Children, Department of Pediatrics, Drexel University College of Medicine, Philadelphia, PA 19134, USA; legido-agustin@cooperhealth.edu

**Keywords:** CDK5RAP3, ER stress, γ-tubulin, microtubule nucleation, UFL1

## Abstract

ER distribution depends on microtubules, and ER homeostasis disturbance activates the unfolded protein response resulting in ER remodeling. CDK5RAP3 (C53) implicated in various signaling pathways interacts with UFM1-protein ligase 1 (UFL1), which mediates the ufmylation of proteins in response to ER stress. Here we find that UFL1 and C53 associate with γ-tubulin ring complex proteins. Knockout of *UFL1* or *C53* in human osteosarcoma cells induces ER stress and boosts centrosomal microtubule nucleation accompanied by γ-tubulin accumulation, microtubule formation, and ER expansion. C53, which is stabilized by UFL1, associates with the centrosome and rescues microtubule nucleation in cells lacking UFL1. Pharmacological induction of ER stress by tunicamycin also leads to increased microtubule nucleation and ER expansion. Furthermore, tunicamycin suppresses the association of C53 with the centrosome. These findings point to a novel mechanism for the relief of ER stress by stimulation of centrosomal microtubule nucleation.

## 1. Introduction

The endoplasmic reticulum consists of a continuous network of membranous sheets and tubules spanning the cytoplasm. It plays critical roles in a wide range of processes, including synthesis, folding, modification, and transport of proteins, synthesis and distribution of lipids, and Ca^2+^ storage. A diverse array of cellular stresses can lead to an imbalance between the protein-folding capacity and protein-folding load [1]. The perturbation of ER homeostasis (ER stress) is ameliorated by triggering signaling cascades of the unfolded protein response (UPR), which engage effector mechanisms leading to homeostasis restoration. These mechanisms include increasing the capacity of the ER, increasing the degradation of ER luminal proteins or upregulation of chaperones and luminal protein modifications, or folding enzymes [2]. In the course of ER stress, the ER undergoes rapid and extensive remodeling hallmarked by the expansion of its lumen and an increase in tubules [3]. In mammalian cells, ER distribution is dependent on microtubules [4].

Microtubules, composed of α- and β-tubulin heterodimers, are highly dynamic and display dynamic instability characterized by altering phases of growth and shrinkage. During interphase, microtubules are mainly nucleated at the centrosome (microtubule organizing centers; MTOC) and radiate toward the cell periphery. γ-Tubulin, a conserved member of the tubulin superfamily, is essential for microtubule nucleation in all eukaryotes [5]. Together with other proteins named γ-tubulin complex proteins (GCPs; GCP2-6), it assembles into a γ-tubulin ring complex (γTuRC), which in mammalian cells effectively catalyzes microtubule nucleation. GCP2-6 each bind directly to γ-tubulin and assemble into the cone-shaped structure of γTuRC [6]. Recent high-resolution structural studies revealed details of γTuRC asymmetric structure [7,8,9]. The γTuRCs typically concentrate at MTOCs such as centrosomes, but they also associate with cellular membranes, where they participate in non-centrosomal microtubule nucleation [10]. The majority of γTuRCs are generally inactive in the cytosol and become active at MTOCs. The mechanisms of γTuRC activation in cells are not fully understood. Current data suggest that γTuRC can be activated by structural rearrangement of γTuRC, phosphorylation, or allosterically by binding to tethering or modulating proteins accumulated in MTOCs [11].

Ubiquitin-fold modifier (UFM1) is a ubiquitin-like posttranslational modifier that targets proteins through a process called ufmylation [12,13]. Conjugation of UFM1 to proteins is mediated by a process analogous to ubiquitination and requires specific activating (E1), conjugating (E2), and ligating (E3) enzymes. Unlike ubiquitination, the modification of proteins by ubiquitin-like modifiers generally serves as a non-proteolysis signal. It regulates various cellular processes by altering the substrate structure, stability, localization, or protein–protein interactions. Ufmylation is reported to regulate multiple cellular processes, including the ER stress response, ribosome function, control of gene expression, DNA damage response, and cell differentiation [14]. Whether ufmylation controls microtubule organization is unknown. E3 UFM1-protein ligase 1 (also known as KIAA0776, RCAD, NLBP, or Maxer; hereafter denoted as UFL1) [12,15,16,17] is mainly located at the cytosolic side of the ER membrane [17], where it is found in complex with DDRGK domain-containing protein 1 (DDRGK1, UFBP1) [18], the first identified substrate of ufmylation [12]. UPR directly controls the ufmylation pathway under ER stress at the transcriptional level [19,20].

Putative tumor suppressor CDK5 regulatory subunit-associated protein 3 (also known as CDK5RAP3, LZAP; hereafter denoted as C53) exerts multiple roles in regulating the cell cycle, DNA damage response, cell survival, cell adherence/invasion, tumorigenesis, and metastasis [21,22,23,24]. Moreover, C53 is also involved in UPR [19]. Several studies have reported the interactions between C53 and UFL1 [15,16,17], and it was suggested that C53 could serve as a UFL1 substrate adaptor [25]. On the other hand, UFL1 regulates the stability of both C53 and DDRGK1 [16]. Reports indicating the involvement of C53 in the modulation of microtubule organization are rare. It has been shown that a peptide from caspase-dependent cleavage of C53 participates in microtubule bundling and rupture of the nuclear envelope in apoptotic cells [26]. In addition, we have demonstrated that C53 forms complexes with nuclear γ-tubulin and that γ-tubulin antagonizes the inhibitory effect of C53 on G_2_/M checkpoint activation by DNA damage [27].

In the present study, we provide evidence that C53, which associates with UFL1 and γTuRC proteins, has an important role in the modulation of centrosomal microtubule nucleation in cells under ER stress. The interaction of ER membranes with newly formed microtubules could promote ER expansion and help restore ER homeostasis.

## 2. Materials and Methods

### 2.1. Reagents

Nocodazole, Cytocholasine B, Ficoll 400, puromycin, geneticin (G418), Hoechst 33342, dimethyl pimelimidate dihydrochloride, and all-*trans*-retinoic acid were from Sigma-Aldrich (St. Louis, MO, USA). Lipofectamine 3000 was from Invitrogen (Carlsbad, CA, USA). Protein A Sepharose CL-4B and Glutathione Sepharose 4 Fast Flow were from GE Healthcare Life Sciences (Chicago, IL, USA). Protease-inhibitor mixture tablets (Complete EDTA-free) were from Roche Molecular Biochemicals (Mannheim, Germany). Restriction enzymes were from New England Biolabs (Ipswich, MA, USA). Oligonucleotides were synthesized by Sigma-Aldrich. Oligopeptides EYHAATRPDYISWGTQ (human γ-tubulin amino acid [aa] sequence 434–449) and EEFATEGTDRKDVFFY (human γ-tubulin aa sequence 38–53) [28] were synthesized by Jerini Peptide Technologies (Berlin, Germany). ER-Tracker Green (BODIPY FL Glibenclamide), AlexaFluor 555-conjugated, and AlexaFluor 647-conjugated wheat germ agglutinin (WGA) were from Thermo Fisher Scientific (Waltham, MA, USA). Purified C-terminally FLAG-tagged C53 and nucleophosmin were from OriGene Technologies (Rockville, MD, USA). Tunicamycin was from Sigma-Aldrich, and 1 mg/mL stock was prepared in DMSO.

### 2.2. Antibodies

Catalog numbers for primary commercial antibodies (Abs) are shown in parentheses. Mouse monoclonal Abs (mAbs) TU-30 (IgG1) and TU-31 (IgG2b) to γ-tubulin aa sequence 434–449 were described previously [29]. Mouse mAbs GTU-88 to γ-tubulin aa sequence 38–53 (IgG1; T6557), TUB2.1 to β-tubulin (IgG1, T4026) and BM-75.2 to α-actinin (IgM, A5044), as well as rabbit Abs to actin (A2066), C53 (HPA022141), calnexin (C4731), DDRGK1 (HPA013373), GAPDH (G9545), GFP (G1544), histone H1.4 (H7665), FLAG (F7425), protein disulfide isomerase (PDI; P7496), and UFL1 (HPA030560; aa sequence 301–389) were from Sigma-Aldrich. Mouse mAbs to GCP4 (IgG1, sc-271876), GCP6 (IgG1, sc-374063), and nucleolin (IgG1, sc-56640), as well as rabbit Abs to SHP-1 (sc-287) and PKCα (sc-208) were from Santa Cruz Biotechnology (Dallas, TX, USA). Rabbit Ab to calcineurin (2614) was from Cell Signaling Technology (Danvers, MA, USA). Mouse mAbs to C53 (IgG1; ab-57817) and DNAdamage-inducible transcript 3 protein (DDIT3; IgG2b, ab11419), as well as rabbit Abs to pericentrin (ab4448) and ODF2 (ab43840) were from Abcam (Cambridge, UK). Mouse mAb to GM130 (IgG1; 610822) was from BD Transduction Laboratories (San Jose, CA, USA), rabbit Ab to GFP (11-476-C100) was from Exbio (Prague, Czech Republic), and rabbit Ab to pericentrin (ABT59) was from EMD-Millipore (La Jolla, CA, USA). Rabbit Ab to α-tubulin (600-401-880) was from Rockland Immunochemicals (Limerick, PA, USA), and rabbit Ab to tRFP was from Evrogen (Moscow, Russia). Mouse mAbs GCP2-01 (IgG2b) and GCP2-02 (IgG1) to GCP2 were described previously [30], as well as mAb to α-tubulin TU-01 [31]. Rabbit Ab to non-muscle myosin heavy chain (BT-561; Biomedical Technologies, Stoughton, MA, USA) and mAb MT-03 to microtubule-associated protein MAP2ab (IgG2b) [32] served as negative controls in the immunoprecipitation experiments. Rabbit Ab to GST was from Dr. Pe. Dráber (Institute of Molecular Genetics, CAS, Prague, Czech Republic).

Anti-mouse and anti-rabbit Abs conjugated with horseradish peroxidase (HRP) were from Promega Biotec (Madison, WI, USA). TrueBlot anti-rabbit IgG HRP was purchased from Rockland Immunochemicals. Anti-mouse Abs conjugated with DyLight 649, DyLight 549, or AlexaFluor 488 and anti-rabbit Abs conjugated with Cy3 or AlexaFluor 488 were from Jackson Immunoresearch Laboratories (West Grove, PA, USA).

Rabbit Ab to human UFL1 sequence 438–793 was prepared by immunizing three rabbits with purified GST-tagged human UFL1 fragment (GST-hUFL1_438-793). Serum with the highest titer to GST-hUFL1_438-793 was partially purified and concentrated by ammonium sulfate precipitation. The precipitate was dissolved and dialyzed against phosphate-buffered saline (PBS). Antigen or GST alone were covalently linked to Glutathione Sepharose 4 Fast Flow by dimethyl pimelimidate dihydrochloride as described [33]. Prepared carriers were thereafter used for affinity isolation of Ab to UFL1. Shortly, a partially purified Ab was first preabsorbed with immobilized GST to remove anti-GST Abs. Then, Ab was bound onto the carrier with GST-hUFL_438-793. The solution of 0.1 M glycine-HCl, pH 2.5, was used for elution, and 0.2 mL fractions were immediately neutralized by adding 20 µL of 1 M Tris-Cl, pH 8.0. Affinity-purified Ab was stained on the blot GST-UFL1, but not GST alone, and detected UFL1 in whole-cell lysates.

### 2.3. Cell Cultures

Human osteogenic sarcoma cell line U-2 OS (U2OS) (Catalog No. ATCC, HTB-96), human glioblastoma cell line T98G (Catalog No. ATCC, CRL-1690), and human cervix adenocarcinoma HeLa S3 (Catalog. No. ATCC, CCL-2.2) were obtained from the American Type Culture Collection (Manassas, VA, USA). Cells were grown at 37 °C in 5% CO_2_ in DMEM supplemented with 10% FCS and antibiotics. P19.X1 cells, a subclone of mouse embryonal carcinoma cells P19, were cultured and subsequently differentiated by incubating cells with 1 μM all-*trans*-retinoic acid for nine days as described [34].

For ER live-cell imaging, cells were transferred to FluoroBrite™ DMEM (Thermo Fisher Scientific) without serum and incubated for 15 min at 37 °C with 1 µM ER-Tracker Green. The staining solution was replaced with a probe-free medium and viewed by fluorescence microscopy. In some cases, cells were treated with 1 µg/mL tunicamycin or carrier (DMSO) for 24 h.

### 2.4. DNA Constructs

To prepare N-terminally EGFP-tagged human UFL1 (*UFL1*; Ref ID: NM_015323.4), the coding sequence was amplified by PCR from pF1KA0776 plasmid (Kasuza DNA Research Institute, Japan), containing the full-length cDNA of human UFL1. The following forward and reverse primers containing sites recognized by *Kpn*I/*Bam*HI restriction endonucleases (underlined) were used: forward 5′-ACGGTACCATGGCGGACGCCT-3′ and reverse 5′-GGTGGATCCTTACTCTTCCGTCACAGATGA-3′. The PCR product was ligated to pEGFP-C1 vector (Clontech Laboratories, Mountain View, CA, USA), resulting in plasmid pEGFP-hUFL1. To prepare C-terminally TagRFP-tagged human UFL1, the coding sequence without a stop codon was amplified by PCR from the pF1KA0776 plasmid. The following primers containing sites recognized by *Nhe*I/*Sal*I were used: forward 5′-ATTGCTAGCAGAACCATGGCGGACGCCT-3′ and reverse 5′-CGCGTCGACCTCTTCCGTCACAGATGATT-3′. The PCR product was ligated to the pCI-TagRFP vector [35], resulting in plasmid phUFL1-TagRFP. To prepare N-terminally GST-tagged full-length UFL1 (aa 1-794), the coding sequence was amplified from the pF1KA0776 plasmid by PCR. The following primers containing sites recognized by *Bam*HI/*Sal*I were used: forward 5′-ATAGGATCCATGGCGGACGCCTGG-3′ and reverse 5′-CGAGTCGACTTACTCTTCCGTCACAGATGATT-3′. The PCR product was ligated into pGEX-6P-1 (Amersham Biosciences, Uppsala, Sweden) using *Bam*HI/*Sal*I restriction sites, resulting in plasmid pGST-hUFL1_1-794. To generate Ab to human UFL1, the GST-tagged C-terminal fragment (aa 438–793) of UFL1 was prepared. The coding sequence was amplified from plasmid pF1KA0776 by PCR using the following primers containing sites recognized by *Bam*HI/*Sal*I: forward 5′-ATAGGATCCGGCAATGCCAGAGAG-3′; reverse 5′-TATCGTCGACTCACTCTTCCGTCACAG-3′. The PCR product was ligated into pGEX-6P-1 using *Bam*HI/*Sal*I restriction sites, resulting in plasmid pGST-hUFL1_438-793. To prepare C-terminally FLAG-tagged human UFL1, the coding sequence was amplified by PCR from the pF1KA0776 plasmid. The following forward and reverse primers containing sites recognized by *Not*I/*Bam*HI were used: forward 5′-TTGCGGCCGCTAAAGGAGATAGAACCATG-3′ and reverse 5′-CGCTGGATCCCTCTTCCGTCACAGATG-3′. The PCR product was ligated to the pFLAG-CMV-5a vector (Sigma-Aldrich), resulting in plasmid phUFL1-FLAG.

To prepare C-terminally EGFP-tagged human CDK5RAP3 (*CDK5RAP3*; Ref ID: NM_176096.3), the coding sequence without stop codon was amplified by PCR from the Myc-DDK-tagged CDK5RAP3 (tv3) plasmid (OriGene Technologies; Catalog No. RC209901). The following primers containing sites recognized by *Nhe*I/*Sal*I were used: forward 5′-TGCTAGCGGAGGAAAGATGGAGGAC-3′ and reverse 5′-TGTCGACCAGAGAGGTTCCCATCAG-3′. The PCR product was ligated into the pCR2.1 vector (Invitrogen) using *Nhe*I/*Sal*I sites, resulting in plasmid pCR-hCDK5RAP3. The complete sequence of CDK5RAP3 without the stop codon was excised from pCR-hCDK5RAP3 by *Nhe*I/*Sal*I and ligated into pEGFP-N3 (Clontech), resulting in plasmid phCDK5RAP3-EGFP. To prepare C-terminally TagRFP-tagged human CDK5RAP3, the coding sequence without stop codon was excised from pCR-hCDK5RAP3 by *Nhe*I/*Sal*I and inserted into pCI-TagRFP [35], resulting in plasmid phCDK5RAP3-TagRFP. Plasmid pGST-hCDK5RAP3 encoding N-terminally GST-tagged full-length human CDK5RAP3 was described previously [27].

To prepare mNeonGreen-tagged lentiviral vector with puromycin resistance, the coding sequence of mNeonGreen was digested out from the mNeonGreen-EB3-7 plasmid (Allele Biotechnology, San Diego, CA, USA) by *Bam*HI/*Not*I. It was thereafter inserted into the pCDH-CMV-MCS-EF1-puro vector (System Biosciences, Palo Alto, CA, USA), resulting in vector mNeonGreen-puro. To prepare C-terminally mNeonGreen-tagged human γ-tubulin 1, the coding sequence was excised from pH3–16 plasmid [28] by *Nhe*I/*Eco*RI and ligated into mNeonGreen-puro, resulting in vector phTUBG1-mNeonGreen-puro. Plasmid pGST-hTUBG1 encoding N-terminally GST-tagged full-length human γ-tubulin-1 was described previously [36].

To prepare control C-terminally TagRFP-tagged mouse SHP-1, the coding sequence was excised from a pmSHP-1-EGFP plasmid (Klebanovych, 2019) by *Eco*RI/*Sal*I and ligated into pCI-Tag-RFP [35], resulting in plasmid pmSHP-1-TagRFP.

CRISPR/Cas9 gene editing [37] was used to disrupt the expression of human UFL1 (Ensembl: ENSG00000014123) or all human CDK5RAP3 variants (Ensembl: ENSG00000108465). Plasmids SpCas9 and pU6-sgRNAnew-III (donated by Dr. R. Malík, Institute of Molecular Genetics, CAS, Prague, Czech Republic) were used for optimal production of Cas9 and single-guide RNA (sgRNA), respectively. The CRISPR tool (https://zlab.bio/guide-design-resources; accessed on 10 November 2017) was used to design the DNA oligonucleotides (for production of sgRNA) that were cloned into *Bsm*BI sites of pU6-sgRNAnew-III. To enrich for cells with disrupted expression of UFL1 or CDK5RAP3, we used the pRR-puro plasmid with a multiple cloning site that encodes a non-functional puromycin resistance cassette [38]. Annealed sense and anti-sense oligonucleotides containing the sequences from the region of interest and overhangs with *Aat*II/*Sac*I restriction sites were ligated into pRR-puro digested with *Aat*II/*Sac*I, resulting in reporter plasmids pRR-hUFL1-puro or pRR-hCDK5RAP3-puro. Co-transfection of the reporter plasmid with the plasmids encoding sgRNAs and Cas9 led to CRISPR-induced double-strand break (DSB) in the reporter plasmid. When the DSB was repaired by homologous recombination, the puromycin resistance was restored.

### 2.5. Generation of UFL1 and C53 Deficient Cell Lines

In order to delete part of the 5′ region of the UFL1 gene containing the canonical and alternative start codons, U2OS cells were transfected with CRISPR/Cas9 vectors (sgRNA#1, sgRNA#2, SpCas9) together with reporter plasmid pRR-hUFL1-puro, for the enrichment of cells not expressing UFL1, by transfection using Lipofectamine 3000 according to the manufacturer’s instructions. The final transfection mixture in a 24-well plate contained 200 ng of sgRNA#1200 ng of sgRNA#2200 ng of reporter plasmid, and 400 ng of SpCas9 in 1 mL of Dulbecco’s modified Eagle’s medium (DMEM) containing 10% FCS, penicillin (100 units/mL), and streptomycin (0.1 mg/mL). The medium was changed after 48 h, and puromycin was added to the final concentration of 2 µg/mL. The stable selection was achieved by culturing cells for 1 wk in the presence of puromycin. The single-cell dilution protocol was used to obtain cell clones that were thereafter analyzed by PCR and immunoblotting. Single-cell clones were expanded, genomic DNA was extracted with the QIAamp DNA Mini Kit (QIAGEN, Gilden, Germany), and deletion in the UFL1 gene was determined by PCR amplification with primers flanking the deleted region: forward 5′-AGGCGCCAATCTTAGACACAG-3′; reverse 5′-CAAAAGCTGCCCTTTTATCTGT-3′. To make sure that all alleles of UFL1 gene were targeted, the PCR amplification was also performed with primers directed to the deleted region: forward: 5′-TCCCTTTAGCTTAGTGATTTGG-3′; reverse 5′-CTAATCTCTTCCCAGGCGTC-3′. Amplified fragments were visualized in 2% agarose gels stained by GelRed Nucleic Acid Gel Stain (Biotium, Fremont, CA, USA). While amplification of short fragments (~570 bp) were detected in UFL1 deficient clones, no amplification was found in control U2OS due to the large size of the deleted region (~7 kb). The PCR fragments were subcloned into the pCR2.1 vector (Invitrogen), and individual colonies were sequenced.

A similar approach was applied to the preparation of cells deficient in CDK5RAP3. In order to delete part of the 5′ region of the GDK5RAP3 gene containing the canonical and alternative start codons, U2OS cells were transfected with CRISPR/Cas9 vectors (sgRNA#1, sgRNA#2, SpCas9) together with reporter plasmid pRR-hCDK5RAP3-puro, for the enrichment of cells not expressing C53. Deletion in the CDK5RAP3 gene was determined by PCR amplification with primers flanking the deleted region: forward 5′-CATGCATCCATCATCCCAG-3′; reverse 5′-TGACATGTGACGTGTGAAACTCT-3′. To make sure that all alleles of the CDK5RAP3 gene were targeted, the PCR amplification was also performed with primers directed to the deleted region: forward: 5′-GCGGAGCCGACTTCATCTT-3′; reverse 5′-TGCAACGTGTTGGGCGATTA-3′. Amplified fragments were visualized in agarose gels. While amplification of short fragments (~700 bp) was detected in CDK5RAP3 deficient clones, no amplification was found in control U2OS due to the large size of the deleted region (~6 kb). The PCR fragments were subcloned into the pCR2.1 vector (Invitrogen), and individual colonies were sequenced.

### 2.6. Generation of Cell Lines Expressing Tagged Proteins

To prepare U2OS cells expressing EGFP-, TagRFP-, or mNeonGreen-tagged proteins, cells were transfected with 2.5 μg DNA per 3-cm tissue culture dish using Lipofectamine 3000 according to the manufacturer’s instructions. After 12 h, the transfection mixture was replaced with a complete fresh medium, and cells were incubated for 48 h. Cells were thereafter incubated for one wk in a complete fresh medium containing G418 at a concentration of 1.2 mg/mL. In the case of phTUBG1-mNeonGreen-puro transfection, cells were incubated in the presence of puromycin at a concentration of 2 µg/mL. For phenotypic rescue experiments, cells expressing TagRFP-tagged proteins were flow-sorted using a BD Influx cell sorter (BD Bioscience, San Jose, CA, USA). TagRFP emission was triggered by 561 nm laser; fluorescence was detected with a 585/29 band-pass filter.

### 2.7. RNA Interference

U2OS cells expressing EB3-mNeonGreen in 6-well plates were transfected with short interfering RNAs (siRNAs) at a final concentration of 10 nM using Lipofectamine RNAi MAX (Invitrogen) according to the manufacturer’s instruction. Cells were harvested 72 h after transfection. The siRNAs that target the regions present in the human DDRGK domain containing 1 (*DDRGK1*, NCBI RefSEq: NM_023935.3), as well as negative control siRNA (Silencer Negative Control #1 siRNA), were purchased from Applied Biosystems (Waltham, MA, USA). The immunoblot analysis revealed that the highest reduction of DDRGK1 was obtained with siRNA ID #s35321; 5′-GAAAATTGGAGCTAAGAAA-3′ (KD1) and siRNA ID #s35322; 5′-CCATAAATCGCATCCAGGA-3′ (KD2).

### 2.8. Real-Time qRT-PCR

Total RNA from cells was isolated with the RNeasy Mini Kit (QIAGEN) and converted to cDNA using the High-Capacity cDNA Reverse Transcription Kit (Applied Biosystems) according to the manufacturer’s protocol. The quantitative PCRs were performed with gene-specific primers specific for human calnexin (*CANX*, Gene ID: 821; primers anneal to all transcript variants), PDI/prolyl 4-hydroxylase subunit beta (*P4HB*, Gene ID: 5034; primers anneal to all transcript variants), CHOP/DNA damage-inducible transcript 3 (*DDIT3*, Gene ID: 1649; primers anneal to all transcript variants except NM_001195057.1), Grp78/BIP/heat shock protein family A (Hsp70) member 5 (*HSPA5*, Gene ID: 3309; primers anneal to all transcript variants), IRE1α/endoplasmic reticulum to nucleus signaling 1 (*ERN1*, Gene ID: 2081; primers anneal to all transcript variants), and ATF6α/activating transcription factor 6 alpha (*ATF6A*, Gene ID: 22926), Cyclophilin A/peptidylprolyl isomerase A (*PPIA*, Gene ID: 5478; primers anneal to all transcript variants) served as an internal control. Primer sequences are summarized in Appendix A. All primers were tested in silico by the Basic Local Alignment Search Tool from the National Center for Biotechnology Information (BLAST NCBI; NIH, Bethesda, MD, USA) to amplify the specific targets. Quantitative PCRs were performed in a LightCycler 480System (Roche, Mannheim, Germany) as described in [39]. Each sample was run in triplicate. The identity of the PCR products was verified by sequencing.

### 2.9. XBP1 mRNA Splicing Assay

An XBP1 mRNA splicing assay was performed as previously described in [40]. Total RNA from control, UFL1_KO, or C53_KO cells was isolated with the RNeasy Mini Kit (QIAGEN), and 1 μg of RNA was converted to cDNA using High-Capacity cDNA Reverse Transcription Kit (Applied Biosystems) according to the manufacturer’s protocol. PCR primers 5′-CGGAAGCCAAGGGGAATGAAG-3′ and 5′-GGATATCAGACTCTGAATC-3′ encompassing the spliced sequences in XBP-1 mRNA were used for the amplification with Combi PPP Master Mix (Top-Bio, Prague, Czech Republic). Amplified fragments were visualized in 3% agarose gel stained by GelRed Nucleic Acid Gel Stain (Biotium).

### 2.10. Preparation of Cell Extracts

Whole-cell lysates for SDS-PAGE were prepared by washing the cells in cold HEPES buffer (50 mM HEPES pH 7.6, 75 mM NaCl, 1 mM MgCl_2,_ and 1 mM EGTA), solubilizing them in hot SDS-sample buffer, and boiling for 5 min. When preparing whole-cell extracts for immunoprecipitation and GST pull-down assays, cells grown on a 9-cm Petri dish (cell confluence 80%) were rinsed in HEPES buffer and extracted (0.7 mL/Petri dish) for 10 min at 4 °C with HEPES buffer supplemented with protease and phosphatase (1 mM Na_3_VO_4_ and 1 mM NaF) inhibitors and 1% NP-40 (extraction buffer). The suspension was then spun down (12,000× *g*, 15 min, 4 °C), and the supernatant was collected. When preparing whole-cell extracts for gel filtration chromatography, cells grown on six 9-cm Petri dishes were scraped into cold HEPES buffer, and pelleted cells were extracted in 1 mL of extraction buffer. The suspension was then spun down as above, and the supernatant was collected.

To prepare the crude membranous fraction, cells from five 9-cm Petri dishes were released by trypsin, washed in HEPES, and mechanically disrupted in 3.5 mL of cold HEPES buffer supplemented with inhibitors using a Dounce homogenizer (disruption efficiency was verified under a microscope). The homogenate was centrifuged at 300× *g* for 5 min (supernatant S1, pellet P1). Post-nuclear supernatant was centrifuged at 1400× *g* for 10 min (supernatant S2, pellet P2). Pelleted material (P2; crude membranous fraction) was solubilized for 5 min at 4 °C with 1.4 mL of extraction buffer, and the suspension was spun down (12,000× *g*, 15 min, 4 °C). The supernatant was collected for immunoprecipitation experiments.

For analysis of the tubulin polymers, cells grown on a 6-well tissue culture plate were rinsed twice in MES buffer (100 mM MES adjusted to pH 6.9 with KOH, 2 mM EGTA, 2 mM MgCl_2_) at 37 °C and then extracted 2 min at 37 °C with MES buffer supplemented with protease and phosphatase inhibitors, 2 M glycerol and 0.2% Triton X-100 (0.5 mL/well). The buffer was carefully removed, and the remaining material containing cytoskeleton with nuclei was resuspended in hot SDS-sample buffer (0.5 mL/well) and boiled for 5 min.

### 2.11. Centrosome Isolation

Centrosomes were prepared as described [41]. Shortly, U2OS cells on ten 15-cm Petri dishes were treated with nocodazole (10 μg/mL) and Cytocholasin B (5 μg/mL) for 90 min to depolymerize microtubules and actin filaments, rapidly washed to remove all salts and lysed in 0.5% NP-40 in low ionic strength buffer. Cell debris and chromatin were spun down (1500× *g*, 5 min, 4 °C), and centrosomes were concentrated onto the 20% (*w*/*w*) Ficoll cushion by centrifugation (25,000× *g*, 20 min, 4 °C). Collected centrosomes were layered onto a discontinuous gradient made up of 5 mL 70%, 3 mL 50%, and 3 mL 40% (*w*/*w*) sucrose in a single thin wall, Ultra-Clear Beckman tube (Cat. No. 344058; Beckman Coulter, Brea, CA, USA). After centrifugation (130,000× *g*, 90 min, 4 °C) in SW 32 Ti rotor (Beckman Coulter), the gradient was fractionated from the bottom, and 0.5 mL fractions were collected. Sucrose density was determined by a refractometer. Centrosome-containing fractions were stored in aliquots in liquid nitrogen. For immunostaining, centrosomes were pelleted onto glass coverslips and fixed in methanol at −20 °C for 5 min.

### 2.12. Gel Filtration Chromatography

Gel filtration was performed using fast protein liquid chromatography (AKTA-FPLC system) on a Superose 6 10/300 GL column or Superose 6 Increase 10/300 GL column (GE Healthcare Life Sciences) as described previously in [27]. The column equilibration and chromatography were performed in HEPES buffer, and 0.5-mL aliquots were collected. Samples for SDS-PAGE were prepared by mixing with 5x concentrated SDS-sample buffe

### 2.13. Immunoprecipitation, GST Pull-Down Assay, Gel Electrophoresis, and Immunoblotting

Immunoprecipitation was performed as previously described in [34]. Extracts were incubated with beads of protein A saturated with mouse mAbs to (i) γ-tubulin (TU-31; IgG2b), (ii) GCP2 (GCP2-01; IgG2b), (iii) MAP2 (MT-03; IgG2b; negative control), or with rabbit Abs to (iv) UFL1_301–389_, (v) UFL1_438–793_, (vi) C53 (Sigma-Aldrich), (vii) (GFP), (viii) non-muscle myosin (negative control), or with (ix) immobilized protein A alone. Antibodies C53, GFP, UFL1_301–389_, and UFL1_438–793_ were used at Ig concentrations 0.5–2.5 µg/mL. Ab to myosin was used at a dilution of 1:100. The mAbs TU-31, GCP2-01, and MT-03, in the form of hybridoma supernatants, were diluted 1:2. To identify proteins interacting with membrane-bound γ-tubulin, 1% NP-40 extract from the microsomal fraction (P_100_) of differentiated P19 cells [36] was incubated with anti-peptide mAb TU-31 immobilized on protein A carrier. After extensive washing, the bound proteins were eluted with immunizing peptide EYHAATRPDYISWGTQ (human γ-tubulin, aa sequence 434–449) [29] as described in [27]. Peptide EEFATEGTDRKDVFFY (human γ-tubulin, aa sequence 38–53) served as a negative control.

Preparation of GST-tagged proteins and pull-down assays with whole-cell extracts were performed as described in [34]. Alternatively, sedimented beads with immobilized GST-γ-tubulin were incubated with FLAG-tagged UFL1, C53 or nucleophosmin at a concentration of 0.5 μg/mL in TBST (10 mM Tris-HCl, pH 7.4, 150 mM NaCl, 0.05% Tween 20). The FLAG-tagged UFL1 was purified using ANTI-FLAG M2 Affinity Gel (Sigma-Aldrich) according to the manufacturer’s protocol.

Gel electrophoresis and immunoblotting were performed using standard protocols. For immunoblotting, mouse mAbs to γ-tubulin (GTU-88), nucleolin, GCP4, β-tubulin, α-actinin, GCP6, DDIT3, and GM130 were diluted 1:10,000, 1:2000, 1:1000, 1:1000, 1:1000, 1:500, 1:500, and 1:250, respectively. Mouse mAbs to α-tubulin (TU-01) and GCP2 (GCP2-02), in the form of spent culture supernatant, were diluted 1:10 and 1:5, respectively. Rabbit Abs to actin, GFP (Sigma-Aldrich), UFL1 (Sigma-Aldrich), C53 (Sigma-Aldrich), tRFP, and calcineurin were diluted 1:10,000, 1:5000, 1:3000, 1:3000, 1:2000, and 1:1000, respectively. Rabbit Abs to PDI, ODF2, FLAG, histone H1.4, pericentrin (EMD-Millipore), and DDRGK1 were diluted 1:50,000, 1: 4000, 1:1000, 1:1000, 1:1000, and 1:500, respectively. Rabbit Abs to calnexin, GAPDH, GST, and SHP-1 were diluted 1:100,000. Secondary anti-mouse and anti-rabbit Abs conjugated with HRP were diluted 1:10,000. TrueBlot anti-rabbit IgG HRP was diluted 1:100,000. The HRP signal was detected with SuperSignal WestPico or Supersignal West Femto Chemiluminescent reagents from Pierce (Rockford, IL, USA) and the LAS 3000 imaging system (Fujifilm, Düsseldorf, Germany). The AIDA image analyzer v5 software (Raytest, Straubenhardt, Germany) was used to quantify signals from the immunoblots.

### 2.14. Mass Spectrometry

Concentrated samples after peptide elution were dissolved in 2× Laemmli sample buffer and separated in 8% SDS-PAGE. Gels were stained with Coomassie Brilliant Blue G-250. The band of interest was excised from the gel, destained, and digested by trypsin. The extracted peptides were analyzed by a MALDI-TOF mass spectrometer (Ultraflex III; Bruker Daltonics, Bremen, Germany) in a mass range of 700–4000 Da and calibrated internally using the monoisotopic [M+H]^+^ ions of trypsin auto-proteolytic fragments (842.51 and 2211.10 Da). Data were processed using FlexAnalysis 3.3 software (Bruker Daltonics) and searched by the in-house Mascot search engine against the SwissProt database subset of all *Mus musculus* proteins.

### 2.15. Evaluation of Cell Growth and FACS Analysis

Cell proliferation was assessed by manual cell counting of control U2OS, UFL1_KO, or C53_KO cells. A total of 2 × 10^5^ cells diluted in culture medium were plated on a 6-cm-diameter Petri dish. Cells were counted at various time intervals from 1 to 4 days. Samples were counted in doublets in a total of five independent experiments. To evaluate changes in cell cycle by FACS analysis, cells were fixed with ice-cold 70% (*v*/*v*) ethanol for 2 h. Fixed cells were washed twice with PBS, collected by centrifugation and resuspended in a staining solution containing 20 μg/mL propidium iodide (Sigma-Aldrich), 100 μg/mL RNAse A (Thermo Fisher Scientific), and 0.1% Triton X-100 in PBS. After 30 min incubation at room temperature, the DNA content was analyzed using a FACSAria IIu flow cytometer (BD Biosciences).

### 2.16. Microtubule Regrowth Experiments

Microtubule regrowth from centrosomes was followed in a nocodazole washout experiment. Cells growing on coverslips were treated with nocodazole at a final concentration of 10 μM for 90 min at 37 °C to depolymerize microtubules. Cells were then washed with PBS precooled to 4 °C (3 times 5 min each) to remove the drug, transferred to complete medium tempered to 28 °C, and microtubule regrowth was allowed for 1–3 min at 28 °C. Cells were after that fixed in formaldehyde and extracted in Triton X-100 (F/Tx) and postfixed in cold methanol (F/Tx/M) as described in [42].

### 2.17. Immunofluorescence Microscopy

Cells were fixed and immunostained as described [42]. Samples were fixed in F/Tx, and for double-label experiments with anti-γ-tubulin Ab, they were postfixed in methanol (F/Tx/M). To visualize TagRFP-tagged proteins, cells were permeabilized with 10 µM digitonin in CHO buffer (25 mM HEPES, 2 mM EGTA, 115 mM CH_3_COOK, 2.5 mM MgCl_2_, 150 mM sucrose, pH 7.4) for 30 s, then fixed with 3% formaldehyde in MSB for 20 min at room temperature and postfixed in methanol at −20 °C for 5 min (D/F/M). Alternatively, TagRFP-tagged proteins were visualized in samples extracted in 0.5% Triton X-100 for 1 min at room temperature, then fixed with 3% formaldehyde in MSB for 20 min at room temperature and postfixed in methanol (Tx/F/M). Centrosomes were fixed in methanol (M). Mouse mAbs to β-tubulin (TUB2.1), C53 (Abcam), and DDIT3 were diluted 1:500, 1:200, and 1:50, respectively. Rabbit Abs to calnexin, PDI, pericentrin, and UFL1_438–793_ were diluted 1:1000, 1:1000, 1:250, and 1:50, respectively. Mouse mAb to γ-tubulin (TU-30), in the form of spent culture supernatant, and rabbit Ab to α-tubulin were diluted 1:10 and 1:100, respectively. Secondary AlexaFluor 488-, DyLight 549-, and DyLight 649-conjugated anti-mouse Abs were diluted 1:200, 1:1000, and 1:1000, respectively. The AlexaFluor 488- and Cy3-conjugated anti-rabbit Abs were diluted 1:200 and 1:1000, respectively. Samples were mounted in MOWIOL 4–88 (Calbiochem, San Diego, CA, USA) or MOWIOL 4–88 supplemented with 4,6-diamidino-2-phenylindole (DAPI; Sigma-Aldrich) and examined with an Olympus AX-70 Provis microscope (Olympus, Hamburg, Germany) equipped with a 60×/1.0 water objective or with a Delta Vision Core system (AppliedPrecision, Issaquah, WA, USA) equipped with a 60×/1.42 oil objective.

To quantify the microtubule regrowth, different areas per sample were taken in both fluorescence channels. The sum of γ-tubulin or α-tubulin immunofluorescence intensities was obtained from nine consecutive frames (0.2 μm steps), with the middle frame chosen with respect to the highest γ-tubulin intensity. Quantification of the microtubule regrowth assay was analyzed automatically in 2-μm regions of interest (ROIs) centered at the centrosomes, marked by γ-tubulin staining, using an in-house written macro for Fiji processing program [43].

### 2.18. Microtubule Nucleation Visualized by Time-Lapse Imaging

For time-lapse imaging, U2OS cells expressing EB3-mNeonGreen were grown on a 35 mm µ-Dish with a polymer coverslip bottom (Ibidi GmbH, Gräfelfing, Germany). Prior to imaging, the medium was replaced for FluoroBrite™ DMEM, supplemented with 25 mM HEPES and 1% FCS, 30 min before imaging. Time-lapse sequences were collected in seven optical slices (0.1 µm steps) for 1 min at 1 s interval with the Andor Dragonfly 503 spinning disc confocal system (Oxford Instruments, Abingdon, UK) equipped with a stage top microscopy incubator (Okolab, Ottaviano, Italy), 488 nm solid-state 150 mW laser, HCX PL APO 63×/1.4 oil objective, and Zyla 4.2 PLUS sCMOS camera. For each experiment, at least 10 cells were imaged (acquisition parameters: 40 µm pinhole size, 15% laser power, 50 ms exposure time, 525/50 nm emission filter). The time-lapse sequences were deconvoluted with Huygens Professional software v. 19.04 (Scientific Volume Imaging, Hilversum, The Netherlands), and maximum intensity projection of z stack was made for each time point in Fiji. Newly nucleated microtubules were detected by manual counting of EB3 comets emanating from the centrosomes.

### 2.19. ER Area Quantification

For ER area quantification in fixed (F/Tx) cells, the ER was visualized by rabbit Ab to calnexin and AlexaFluor 488-conjugated anti-rabbit Ab. Samples were then incubated with AlexaFluor 555-conjugated WGA (5 μg/mL) for 30 min at room temperature to delineate cell boundary and mounted in MOWIOL 4-88 with DAPI to mark nucleus. Preparations were examined with the Andor Dragonfly 503 spinning disc confocal microscope (0.13 μm steps) equipped with HCX PL APO 63×/1.4 oil objective and Zyla 4.2 PLUS sCMOS camera. Images were deconvoluted with Huygens Professional software, and the ER area coefficient (area occupied by ER/area free of ER) was calculated from calnexin fluorescence intensity in the cytoplasm of individual cells using an in-house written macro (Appendix A) for Fiji processing program. AlexaFluor 647-conjugated WGA was used when quantification was performed in phenotypic rescue experiments with C53-TagRFP.

For ER area quantification during live-cell imaging, cells were transferred to FluoroBrite™ DMEM without serum and incubated for 15 min at 37 °C with 1 µM ER-Tracker Green, AlexaFluor 555-conjugated WGA (5 μg/mL) and Hoechst 33342 (0.5 μg/mL). After washing in FluoroBrite™ DMEM, the preparations were examined with the Andor Dragonfly 503 spinning disc confocal microscope (0.5 μm steps) equipped with HC PL APO 63×/1.2 water objective, and iXon Ultra 888 EMCCD camera.

### 2.20. Statistical Analysis

A minimum of three independent experiments was analyzed for each quantification. The counts of individual data points were indicated in the figure legends. All data were presented as mean ± SD. Significance was tested using a two-tailed, unpaired Student’s *t*-test or one-way ANOVA followed by a Sidak’s or Dunnett’s post hoc test using Prism 8 software (GraphPad Software, San Diego, CA, USA). The used test are indicated in the figure legends. For all analyses, *p*-values were represented as follows: *, *p* < 0.05; **, *p* < 0.01; ***, *p* < 0.001; ****, *p* < 0.0001.

## 3. Results

### 3.1. Identification of UFL1 as γ-Tubulin Interactor

We have previously reported the intrinsic association of γ-tubulin with cellular membranes in mouse P19 embryonal carcinoma cells undergoing neuronal differentiation [36]. To identify the potential interacting partners for membrane-bound γ-tubulin, we performed immunoprecipitation experiments with anti-peptide mAb to γ-tubulin and extracts from the microsomal fraction of differentiated P19 cells. The bound proteins were eluted with the peptide used for immunization or with a negative control peptide. In repeated experiments, a 90-kDa protein was specifically eluted with the immunization peptide but not with the control peptide (Appendix A). The protein was subjected to MALDI/MS fingerprint analysis and identified as UFL1 (UniProtKB identifier O94874) (Appendix A). UFL1 mainly associates with the endoplasmic reticulum (ER) membranes [12,17] and interacts with C53 [15,16]. We showed that C53 is present in γ-tubulin immunocomplexes from the nuclear fraction of human HeLa S3 cells [27]. Collectively these data indicate the existence of protein complexes comprising UFL1, C53, and γ-tubulin.

### 3.2. UFL1 and C53 Associate with γTuRC Proteins

To ascertain whether membranous UFL1 associates both with C53 and γTuRC proteins in different cell types, we first performed immunoprecipitation experiments with extracts from the crude membranous fraction (P2) from human osteosarcoma U2OS cells. The fraction contained UFL1, C53, γ-tubulin, GCP2, and calnexin (a marker of ER) but was devoid of α-tubulin, GM130, and histone H1.4 representing, cytosolic, Golgi apparatus, and nuclear proteins, respectively (Appendix A). Using two anti-UFL1 Abs recognizing epitopes in distinct UFL1 aa sequence regions (301–389 [UFL1_301–389_] and 438–793 [UFL1_438–793_]) and Abs to C53, γ-tubulin, and GCP2 for reciprocal precipitations, we revealed association of UFL1 and C53 with γTuRC proteins but not with calcineurin, serving as a negative control. As expected, we also verified the association of C53 with UFL1 (Figure 1A and Appendix A). Rabbit anti-myosin and mouse anti-MAP2 (IgG2b) Abs served as isotype controls for immunoprecipitation from membranous fractions (Figure 1B). When we performed immunoprecipitation experiments with membranous fractions from glioblastoma T98G cells, we likewise detected complexes comprising UFL1, C53, and γ-tubulin (Appendix A), indicating that these protein complexes are not limited to U2OS cells.

### 3.3. Association of Exogenous UFL1 and C53 with γTuRC Proteins and Centrosome

To independently validate the interaction of UFL1 and C53 with γTuRC proteins, we performed immunoprecipitation experiments from cells expressing EGFP-tagged UFL1 or C53 and control cells expressing EGFP alone. The Ab to GFP co-precipitated γ-tubulin, GCP2, GCP4, GCP6, and C53 from the cells expressing EGFP-UFL1 or C53-EGFP (Appendix A). Reciprocal precipitation with Ab to γ-tubulin confirmed the interaction of EGFP-tagged UFL1 or EGFP-tagged C53 with γ-tubulin (Appendix A). On the other hand, the Ab to GFP did not co-precipitate UFL1, C53, γ-tubulin, GCP2, or GCP4 from control cells expressing EGFP alone. Similarly, Ab to γ-tubulin did not co-precipitate EGFP from these cells (Appendix A).

To further verify the interaction of UFL1 and C53 with γTuRC proteins, we performed pull-down assays with GST-tagged UFL1, C53, or γ-tubulin. The experiments revealed that γ-tubulin and GCP2 are bound to GST-UFL1 or GST-C53 but not to GST alone (Appendix A). Similarly, UFL1 and C53 are bound to GST-tagged γ-tubulin but not to GST alone (Appendix A). The negative control protein (calcineurin) did not bind to GST-fusion proteins, though the amounts of immobilized GST fusion proteins were comparable, as evidenced by staining with Ab to GST (Appendix A). Collectively, these data show that exogenous UFL1 and C53 form complexes with γTuRC proteins as well.

To determine whether UFL1 and C53 interact directly with γ-tubulin, the pull-down assay was performed with GST-γ-tubulin and purified FLAG-tagged UFL1, C53, or nucleophosmin (NPM1, negative control). C53 was clearly bound to γ-tubulin, in contrast to UFL1. A very weak signal for UFL1 was detectable only on overexposed blots. NPM1 did not bind to GST-γ-tubulin, and UFL1, C53, or NPM1 did not bind to GST alone. The amounts of immobilized GST fusion proteins present in each pull-down were similar (Appendix A). These data suggest that C53 may be able to bind γ-tubulin directly.

As γTuRCs are essential for normal microtubule nucleation from centrosomes, we tested whether UFL1 or C53 localize to the centrosome. However, using immunofluorescence microscopy with a limited number of commercially available Abs to UFL1 and C53, we failed to localize these proteins on the centrosome. We, therefore, expressed TagRFP-tagged UFL1, C53, or protein tyrosine phosphatase SHP-1 (negative control) in U2OS cells. Tagged proteins were expressed in cells in comparable amounts (Appendix A). On fixed cells, C53-TagRFP localized to interphase centrosomes, cytosol, and nuclei (Figure 2Aa–f), while UFL1-TagRFP was found only in the cytosol (Figure 2Ag–i), similarly to control SHP-1-TagRFP (Appendix A). C53-TagRFP was also clearly detected on centrosomes in living cells (Appendix A), in contrast to UFL1-TagRFP.

To independently confirm the association of C53 with centrosomes, we performed immunoblot analysis of centrosomes isolated from U2OS cells by gradient centrifugation. Immunostaining of purified centrosomes with Abs to γ-tubulin and pericentrin is shown in Appendix A. We found C53 in fractions containing centrosomal marker proteins pericentrin, CDK5RAP2, outer dense fiber 2 (ODF2), and γ-tubulin. On the other hand, in the same fractions, we did not detect either UFL1 or PKCα, histone H1.4, and calnexin representing cytosolic, nuclear, and ER proteins, respectively (Figure 2B). These data document that although UFL1 and C53 interact with each other and form complexes with γTuRC proteins, only C53 associates with centrosomes.

### 3.4. Preparation and Characterization of Cell Lines Lacking UFL1 or C53

To evaluate the possible effect of UFL1 and C53 on microtubule nucleation, we prepared U2OS cell lines lacking UFL1 or C53. For that, we took advantage of CRISPR/Cas9 editing. A schematic diagram of the human *UFL1* gene with sites targeted by sgRNA#1 and sgRNA#2, enabling efficient deletion of all UFL1 isoforms, is shown in Appendix A. We established three independent cell lines (denoted UFL1_KO1, UFL1_KO2, and UFL1_KO3) that have deletions in the targeted region (Appendix A) and undetectable UFL1 in immunoblotting (Appendix A). When compared to control cells, a radical decrease in UFL1 immunofluorescence staining was observed in UFL1_KO cells (Appendix A). If not mentioned otherwise, the following results are based on UFL1_KO1 cells (abbreviated UFL1_KO).

We used the same approach to prepare U2OS cells lacking C53. A schematic diagram of the human *CDK5RAP3* gene with sites targeted by sgRNAs for efficient deletion of all C53 isoforms is shown in Appendix A. We established three independent cell lines (denoted C53_KO1, C53_KO2, and C53_KO3) that have deletions in the targeted region (Appendix A) and undetectable C53 in immunoblotting (Appendix A). When compared to control cells, a substantial decrease in C53 immunofluorescence staining was observed in C53_KO cells (Appendix A). If not mentioned otherwise, the following results are based on C53_KO1 cells (abbreviated C53_KO).

To assess the effect of UFL1 or C53 deletion on cell division, we determined cell growth in control and UFL1_KO or C53_KO cells. Compared with control cells, the number of viable cells decreased significantly in both UFL1-KO and C53_KO, but proliferation was more impaired in UFL1_KO cells (Appendix A). Analysis of asynchronous cell cultures by FACS analysis revealed a trend toward more cells in the G1 phase and fewer in G2/M for UFL1_ KO and C53_ KO cell lines (Appendix A).

Quantitative immunoblot analysis showed that the deletion of UFL1 resulted in a substantial reduction of C53 and DDRGK1. On the other hand, the deletion of C53 only resulted in a moderate decrease in UFL1 and DDRGK1. In cells lacking UFL1, the amount of C53 and DDRGK1 dropped to ~10% and ~40% of the wild-type level, respectively. In cells lacking C53, the amount of UFL1 and DDRGK1 decreased to ~75% and ~80% of the wild-type level, respectively. Deletion of UFL1 or C53 did not affect the expression of γ-tubulin or GCP2 (Figure 3A). The size distribution of UFL1 in control and C53_KO whole-cell extracts was comparable, suggesting that the formation of large UFL1 complexes is not exclusively dependent on the presence of C53 (Appendix A).

Experiments on knockout mouse models revealed that the deletion of UFL1 [44,45] or C53 [25] triggered ER stress and activated the UPR. Following initiation of the UPR, which utilises three distinct signalling pathways (IRE1, PERK, and ATF6), protein translation is arrested to allow mRNA translation of UPR proteins, which in turn induce translation of ER chaperones to upregulate protein folding [14]. qRT-PCR analysis of genes involved in the UPR revealed that deletion of UFL1 resulted in significant upregulation of the ER stress sensors IRE1α (*ERN1*) and ATF6α (*ATF6A*), as well as the downstream target Grp78/BIP (*HSPA5*) and the downstream target CHOP (*DDIT3*) of the PERK stress sensor. Deletion of C53 resulted in upregulation of IRE1α and Grp78/BIP (Appendix A). In addition, XBP1 mRNA splicing assay showed that the spliced variant XBP1 (XBP1s) involved in IRE1α signalling was clearly detectable in both UFL1_KO and C53_KO cells (Appendix A). These results document the UPR in the prepared cell lines and suggest that the PERK branch may not be activated in cells lacking C53, as also previously reported for intestinal cells [46].

One of the characteristic features of stressed cells is the expansion of the ER network [47]. To characterize the prepared cell lines with respect to ER stress, we visualized ER in living cells by cell-permeant ER-Tracker. Substantial changes in ER distribution were observed both in UFL1_KO and C53_KO cells. In control cells, the ER tubules at the cell periphery were sparse (Figure 3Ba,a’). In UFL1_KO cells, prominent formation of ER tubules was observed, suggesting the expansion of the ER network (Figure 3Bb,b’). The ER expansion in the C53_KO cell was also present but less prominent than in UFL1_KO cells (Figure 3Bc,c’). For quantification of ER area during live-cell imaging, cells were incubated with ER-Tracker Green, AlexaFluor 555-conjugated WGA to delineate cell border, and Hoechst 33342 to label the nucleus. Calculation of the ER area coefficient (area occupied by ER/area free of ER) confirmed significant ER expansion in both UFL1_ KO and C53_ KO cells (Figure 3C).

Collectively these findings point to the vital role of UFL1 and C53 in ER homeostasis. We confirmed that UFL1 efficiently regulates the C53 protein level. This is in agreement with previous reports documenting the stabilization effect of UFL1 on C53 via suppression of C53 ubiquitination and proteasome degradation [15,16].

### 3.5. UFL1 or C53 Deficiency Increase Centrosomal Microtubule Nucleation

To unravel the effect of UFL1 and C53 deficiency on microtubule nucleation, we followed microtubule regrowth from U2OS centrosomes, which represent the major nucleation centers in interphase cells. Microtubule regrowth in nocodazole-washout experiments in control, UFL1_KO, and C53_KO cells were performed as previously described [48]. The extent of microtubule regrowth could be modulated by mechanisms regulating either microtubule nucleation or microtubule dynamics. It was reported previously that a delay in microtubule regrowth is associated with defects in microtubule nucleation [49]. We measured α-tubulin and γ-tubulin immunofluorescence signals in a 2.0 µm circular ROI, 2 min after nocodazole-washout in UFL1- or C53-deficient cells and controls. When compared with control cells, an increase in microtubule regrowth was observed both in UFL1_KO1 (Figure 4A) and C53_KO1 (Figure 4D) cells. Quantification of γ-tubulin immunofluorescence revealed that the amount of γ-tubulin in centrosomes increased in both UFL1_KO1 (Figure 4B) and C53_KO1 (Figure 4E) cells. Typical staining of α-tubulin and γ-tubulin in control, UFL1_KO1, and C53_KO1 cells is shown in Figure 4C,F. When microtubule regrowth experiments were performed with the other knockout cell lines (UFL1_KO2, UFL1_KO3 or C53_KO2, C53_KO3), similar results were obtained. While the amount of centrosomal γ-tubulin increased in the cells lacking UFL1 or C53, the amount of centrosomal pericentrin was not affected, indicating that general pericentriolar matrix integrity was preserved (Appendix A). To exclude the possibility that enhanced microtubule nucleation reflects an increase in nocodazole-resistant microtubules, we fixed cells in the presence of nocodazole. The microscopic analysis did not reveal enrichment of nocodazole-resistant microtubules in UFL1_KO or C53_KO cells. The centrosome number was also unchanged when compared with control cells (Appendix A).

To independently evaluate the role of UFL1 and C53 in microtubule nucleation, we performed time-lapse imaging in cells expressing mNeonGreen-tagged microtubule end-binding protein 3 (EB3), decorating plus ends of the growing microtubules, and counted the number of EB3 comets leaving the centrosomes per unit time (nucleation rate) [48,49]. When compared to control cells, the nucleation rate increased in cells lacking UFL1 (Figure 4G). A similar effect was observed after the deletion of C53 (Figure 4H). A comparison of a single frame or 10-frame projections from control and UFL1_KO or C53_KO cells is shown in Appendix A. These live-imaging data correspond to the results obtained by measuring the α-tubulin signal during the microtubule regrowth experiment. We also quantified polymerized tubulin in control and UFL1_KO or C53_KO cells and found significantly more microtubules in deficient cells (Appendix A). This suggests that UFL1 or C53 deficient cells have more microtubules at a steady state. To determine whether DDRGK1 plays a role in microtubule nucleation, we efficiently depleted DDRGK1 by siRNAs in U2OS cells expressing mNeonGreen-tagged EB3 (Appendix A). Using time-lapse imaging, we did not detect any difference between nucleation rates in control and DDRGK1-depleted cells (Appendix A). This suggests that DDRGK1 is not involved in the regulation of centrosomal microtubule nucleation.

To verify the specificity of observed changes in microtubule nucleation, we performed rescue experiments by expressing UFL1-TagRFP or TagRFP alone in UFL1_KO cells. The introduction of UFL1-TagRFP into UFL1-deficient cells restored UFL1 expression and enhanced the expression of C53 (Appendix A). While the introduction of UFL1-TagRFP into UFL1_KO cells decreased the microtubule regrowth to that in control cells, the expression of TagRFP failed to do so (Appendix A). Correspondingly, the amount of centrosomal γ-tubulin decreased after the introduction of UFL1-TagRFP into UFL1-KO cells, whereas it remained elevated in UFL1_KO cells expressing TagRFP alone (Appendix A). A similar set of phenotypic rescue experiments was performed in C53_KO cells. C53-TagRFP efficiently restored the C53 level in deficient cells (Appendix A), and microtubule regrowth was restored to that in control cells (Appendix A). In addition, the amount of centrosomal γ-tubulin decreased after introducing C53-TagRFP to deficient cells (Appendix A).

Altogether, these data indicate that UFL1 and C53, but not DDRGK1, negatively regulate microtubule nucleation from the interphase centrosome by influencing centrosomal γ-tubulin/γTuRCs levels.

### 3.6. Enhancement of Microtubule Nucleation in Cells under ER Stress

As live-cell imaging of ER in control, UFL1_KO, and C53_KO cells showed the generation of ER stress in cells lacking UFL1 and C53 (Figure 3B,C), we evaluated changes in the expression and cellular distribution of chaperone calnexin and protein disulfide-isomerase (PDI), well-established markers of UPR, facilitating protein folding [1]. Densitometric analysis of immunoblotting experiments revealed a significantly increased expression of calnexin and PDI in UFL1_KO cells. These UPR markers also increased in C53_KO cells, albeit PDI only moderately (Figure 5A). Immunofluorescence microscopy verified increased expression of calnexin (Figure 5Be,i) and PDI (Appendix A) in cells lacking UFL1 or C53. When compared with control cells, immunostaining for calnexin spread to the cell periphery delineated by ends of microtubules both in UFL1_KO and C53_KO cells, as shown in magnified regions (Figure 5Bg,k). To quantify the ER area in fixed cells, samples were immunostained for calnexin to mark ER, WGA conjugated with AlexaFluor 555 to delineate cell boundary and DAPI to mark the cell nucleus. Calculation of ER area coefficient revealed a significant ER expansion in both UFL1_KO and C53_KO cells, though in the case of C53_KO cells was less prominent (Figure 5C). Similarly, the PDI immunostaining also expanded to the cell periphery in cells lacking UFL1 or C53 (Appendix A).

To test whether C53 can rescue increased microtubule nucleation and ER expansion phenotypes in UFL1_KO cells, C53-TagRFP or TagRFP alone were expressed in UFL1_KO cells. C53-TagRFP was efficiently expressed in UFL1_KO cells characterized by a low amount of C53 (Figure 3A and Figure 6A). In these cells, both microtubule regrowth (Figure 6B) and the amount of centrosomal γ-tubulin (Figure 6C) decreased to the levels in control cells. Densitometric analysis of immunoblots revealed a partial decrease in calnexin protein level in UFL1_KO cells expressing C53-TagRFP (Figure 6D), indicating that C53 partly rescued ER stress induced in UFL1-KO cells. Interestingly, C53 also rescued ER expansion in UFL1_KO cells, as shown by staining for calnexin (Figure 6Ea–c). The edge of the cell was delineated by AlexaFluor 647-conjugated WGA (Figure 6Ed–f). Quantification revealed a significant decrease in ER area in UFL1_KO cells expressing C53-TagRFP compared to UFL1_KO cells expressing TagRFP alone (Figure 6F). These data suggest that the modulation of microtubule nucleation and ER distribution by UFL1 likely occurs by reducing the amount of centrosomal C53.

Microtubules are known to regulate ER homeostasis, with ER dynamics tightly linked to the dynamics of microtubules. This is particularly important during ER stress, as the expansion of the ER is one of the relief mechanisms [3]. To evaluate whether the expansion of ER in stressed cells could be potentially promoted by *de novo* microtubule nucleation, we pretreated cells with tunicamycin, a potent inhibitor of protein glycosylation and ER stress activator [50]. Immunofluorescence microscopy in tunicamycin-treated cells showed expansion of ER, as documented by staining of live cells with ER-Tracker (Figure 7Aa,b), and by staining fixed cells with Abs to calnexin (Figure 7Ac,d) and PDI (Figure 7Ae,f). Moreover, DNA damage-inducible transcript 3 (DDIT3), representing another UPR marker [1], accumulated in nuclei after tunicamycin treatment (Figure 7Ag,h). Increased expression of calnexin, PDI, and DDIT3 after tunicamycin treatment was confirmed by immunoblotting (Appendix A). Changes in the protein level corresponded to changes in the transcript level (Appendix A). Quantification of ER area in samples marked by ER-Tracker revealed the significant extension of ER in tunicamycin-treated cells (Figure 7B). Similar results were obtained when ER was stained by Ab to calnexin (Figure 7C). Interestingly, we detected increased microtubule regrowth (Figure 7D) and centrosomal γ-tubulin accumulation (Figure 7E) in tunicamycin-treated cells. Correspondingly, the nucleation rate also increased in treated cells (Figure 7F,G). The centrosome number in tunicamycin treated cells was unchanged. When C53_KO cells overexpressing C53-TagRFP were pretreated with tunicamycin, fluorescence microscopy revealed that the overall C53-TagRFP signal decreased, and its association with the centrosome was suppressed (Figure 8A). Immunoblot analysis showed that while the amount of C53-TagRFP decreased in tunicamycin-treated cells, protein levels of UFL1 or γ-tubulin were similar in control and drug-treated cells (Figure 8B). The exogenous pool of C53-TagRFP not stabilized by UFL1 might be more prone to degradation after tunicamycin treatment as described for other proteins [51]. No changes in fluorescence staining intensity were observed in C53_KO cells expressing TagRFP alone and treated or not with tunicamycin (Figure 8C). Immunoblot analysis also did not reveal changes in the amount of TagRFP (Figure 8D), ruling out the possibility that tunicamycin affects the expression of tagged proteins.

Pretreatment of wild-type cells with tunicamycin resulted in the lower amount of C53 associated with the P1 fraction, comprising centrosomes and nuclei with connected membranes. On the other hand, a comparable amount of pericentrin was found in P1 fractions of control and drug-treated cells (Figure 8E). To test whether tunicamycin affects the association of C53 with centrosomes, we isolated centrosomes from control and tunicamycin-treated cells. Less C53 is associated with centrosomes from tunicamycin-treated cells. (Figure 8F). To determine whether UFL1 has an effect on microtubule nucleation and C53 levels in tunicamycin-treated cells, we first compared nucleation in cells overexpressing UFL1-TagRFP or TagRFP alone (control). Using time-lapse imaging, we found comparable nucleation rates as documented in Appendix A. Endogenous C53 levels were not affected by overexpression of UFL1-TagRFP (Appendix A) or treatment with tunicamycin (Appendix A). Overexpression of UFL1-TagRFP also did not alter microtubule nucleation in tunicamycin-treated cells (Appendix A).

Collectively, both genetically and pharmacologically induced ER stress increases centrosomal microtubule nucleation and ER expansion by modulating C53 subcellular localization. These data suggest that C53 represents the mechanistic link between the ER stress–response machinery and the microtubule network reorganization.

## 4. Discussion

The ER distribution is dependent on microtubules. Although ER expansion is characteristic of cells under ER stress [2], the molecular mechanisms of microtubule regulation under ER stress conditions are not fully understood. UFL1 and its adaptor C53 are essential for ER homeostasis [14], and their deletions generate ER stress [25]. We used well-adherent U2OS cells suitable for analysis of microtubule nucleation from interphase centrosomes to evaluate microtubule organization in cells lacking UFL1 and C53. We report on UFL1 and C53 association with γTuRC proteins and their involvement in the regulation of centrosomal microtubule nucleation. Increased microtubule nucleation in cells under ER stress likely facilitates ER expansion.

### 4.1. Interaction of UFL1 and C53 with γTuRC Proteins

Several lines of evidence support the conclusion that UFL1 and C53 (UFL1/C53) associate with γTuRC proteins. First, UFL1 was identified by MALDI/MS fingerprinting analysis after immunoprecipitation with anti-peptide mAb to γ-tubulin and elution of bound proteins with the immunizing peptide. Second, reciprocal immunoprecipitations confirmed the formation of complexes containing UFL1, C53, γ-tubulin, and GCPs. Third, separation of cell extracts by gel filtration revealed co-distribution of UFL1, C53, γ-tubulin, and GCP2 in high molecular weight fractions, in which both UFL1 and C53 co-precipitated with γ-tubulin. Fourth, γ-tubulin and GCPs associated with EGFP-tagged UFL1 or C53. Fifth, pull-down assays with whole-cell extracts showed that γ-tubulin and GCP2 bind to GST-tagged UFL1 or C53 and that both UFL1 and C53 interact with GST-γ-tubulin. Finally, a direct interaction between GST-γ-tubulin and C53- FLAG was detected. The interaction of UFL1/C53 with γ-tubulin was also observed in cells of different tissue origins. The association was found in cells of human osteogenic sarcoma (U2OS), glioblastoma (T98G), and cervical adenocarcinoma (HeLa S3). It has been previously reported that C53 forms complexes with nuclear γ-tubulin [27], but it was unclear whether C53 can associate with γ-tubulin in the other cell parts and whether its complexes include GCPs essential for γ-TuRC-dependent microtubule nucleation. The presented results suggest that multiprotein complexes containing UFL1/C53 and γTuRC proteins occur at different cellular locations in various cell types.

UFL1 directly interacts with C53 [15,16], which serves as a substrate adaptor for UFL1 [25]. Both UFL1 and C53 are largely ER-associated proteins [12,16,17]. Reciprocal precipitation of these proteins from membranous fractions was therefore expected. On the other hand, the association of UFL1/C53 with membrane-bound γTuRC proteins was surprising. To our knowledge, such interaction was not reported. We have previously shown that γ-tubulin is associated with detergent-resistant membranes in mouse embryonic carcinoma P19 cells induced to neuronal differentiation [36] and the human brain [39]. In this context, it should be noted that γ-tubulin is essential for microtubule nucleation from the Golgi membranes [52]. γ-Tubulin was also found on recycling endosomes [53] and mitochondrial membranes [39]. It was reported that UFL1 regulates mitochondrial mass [45]. Since ER and mitochondria are known to cross-talk at membrane contact sites, deciphering the role of UFL1/C53 in the regulation of microtubule nucleation from membrane-bound γ-tubulin complexes warrants further investigation.

### 4.2. Microtubule Nucleation in Cells Lacking UFL1 or C53

Although both UFL1 and C53 interacted with γTuRC proteins, tagged fluorescent proteins revealed a centrosomal association in fixed and living cells only in the case of C53. Moreover, C53 but not UFL1 is associated with isolated centrosomes. Centrosomal localization of C53 was previously reported by immunofluorescence microscopy [22]. We failed to localize C53 to centrosomes by microscopy using a panel of commercial Abs under various fixation conditions. Such differences in localization could reflect the exposure of epitopes for the used Abs.

We prepared, using CRISPR/Cas 9 gene editing, cell lines lacking either UFL1 or C53. Cell growth inhibition and activation of UPR were characteristic features of UFL1_KO cells. A similar tendency, but less prominent, was also observed in the case of C53_KO cells. The absence of UFL1 resulted in a substantial reduction of C53 and DDRGK1. On the other hand, only a moderate decrease in the amount of UFL1 and DDRGK1 was observed in cells lacking C53 (Figure 3A). It was reported that UFL1 stabilize C53 by inhibiting its ubiquitination [15] and subsequent proteasome system-mediated protein degradation [16]. UFL1 thus plays a vital role in the regulation of the C53 protein level.

Our data demonstrate that UFL1 and C53 represent negative regulators of microtubule nucleation from interphase centrosomes. The deletion of both UFL1 and C53 increased centrosomal microtubule nucleation. Nevertheless, the loss of UFL1 also leads to a substantial reduction of the C53 level. The rescue of microtubule nucleation phenotype in UFL1_KO cells by C53 (Figure 6A–C), therefore, strongly suggests that C53 is a novel regulator of centrosomal microtubule nucleation. The regulatory role of UFL1 in microtubule nucleation could thus be indirect through modulation of the centrosomal C53 amount. Although complexes of C53 and UFM1 were reported [18], direct ufmylation of C53 was so far not demonstrated [25]. Our results from rescue experiments indicate that putative C53 ufmylation by UFL1 is not required for the modulation of centrosomal microtubule nucleation in UFL1-KO cells.

Microtubule nucleation at the centrosome occurs from γTuRCs located in the pericentriolar material [54]. We, therefore, examined whether UFL1 and C53 regulate microtubule nucleation by affecting the centrosomal γ-tubulin levels. Our data suggest that in wild-type cells, both proteins suppress γ-tubulin accumulation at the centrosome. On the other hand, no changes in the amount of pericentrin were detected, showing that the general pericentriolar matrix integrity is not affected by C53 or UFL1 depletion. Altogether, these data indicate that the regulatory roles of UFL1 and C53 are conveyed by γ-tubulin/γTuRC accumulation on centrosomes. Such a regulatory mechanism of microtubule nucleation is not unique only for UFL1/C53. It has been reported that androgen and Src signaling, which leads to the activation of the ERK, regulates microtubule nucleation by promoting the accumulation of γ-tubulin at the centrosome [49]. Modulation of γ-tubulin accumulation in centrosomes was also shown for GIT1/βPIX signaling proteins and PAK1 kinase [48], and for tyrosine phosphatase SHP-1 [43].

### 4.3. Regulatory Mechanisms by Which UFL1/C53 Can Control Microtubule Nucleation

C53 binds to multiple targets but has no enzymatic domain or other well-characterized functional motifs, suggesting that it may exert its activity through interaction with other proteins. Recently, C53 was identified as a highly conserved regulator of ER stress-induced ER-phagy [55]. Interestingly, accumulating evidence suggests that C53 is also implicated in the regulation of protein phosphorylation. It was reported that C53 was directly bound to protein serine/threonine-protein phosphatase 1D (PP2Cδ) and promoted its phosphatase activity toward several C53 targets [56]. Phosphorylation sites were identified in building components of γTuRC [54]. Many phosphorylation sites were also identified in various γ-TuRC tethering proteins. Several of these sites were characterized and shown to stimulate γTuRC assembly, recruitment, or activation [11]. On the other hand, phosphorylation can also negatively regulate γTuRC recruitment and activity, as hyperphosphorylation of Mto2 in fission yeast leads to the inactivation of γTuRCs at non-spindle pole body sites [57]. The actions of kinases and phosphatases have to be balanced to finely tune microtubule nucleation events during the cell cycle and in response to stress conditions. The activation of phosphatase(s) by C53 might maintain a low level of phosphorylated γTuRCs or TuRC-tethering proteins, resulting in the attenuation of microtubule nucleation.

Although the regulation of microtubule nucleation in UFL1_KO cells could be explained by a low amount of C53, one cannot exclude the possibility that UFL1 affects microtubule nucleation independently of C53. It was reported that the UFM1 cascade in *Drosophila* alters the level of phosphorylation on tyrosine-15 of Cdk1 (pY15-Cdk1), which serves as an inhibitor of G2/M transition. In cells lacking UFL1, the level of pY15-Cdk1 was significantly reduced, indicating increased activity of Cdk1 [58]. Sequential phosphorylation of NEDD1 by Cdk1 and Plk1 is required for targeting γTuRCs to centrosomes [59].

### 4.4. Enhanced Microtubule Nucleation in Cells under ER Stress

One of the characteristic features of U2OS cells without UFL1 is the generation of ER stress, which was previously reported in mouse UFL1 knockout models for bone marrow cells [45] and cardiomyocytes [44]. Similarly, U2OS cells lacking C53 are under ER stress, which was also recently reported in the mouse C53 knockout model for hepatocytes [25]. When compared to control cells, where ER was concentrated around the nucleus, the UFL1_KO or C53_KO cells had an expanded ER network. This was clearly evident by visualizing ER with ER-Tracker or by immunostaining for calnexin or PDI, in which the protein level increased in cells lacking UFL1 or C53. Expansion of ER was already described using staining with Ab to PDI after partial depletion of UFL1 and C53 by RNAi [19]. Expansion of ER was also clearly observable in cells treated with tunicamycin, a pharmacological inducer of ER stress [50].

In mammalian cells, the ER network rearrangements strongly depend on the interactions with dynamic microtubules. This is very important during ER stress, as the expansion of ER alleviates ER stress [3,47]. There are four distinct mechanisms of how ER can be rearranged with the help of microtubules. ER tubules can be pulled out of the existing ER membranes by associating with motor proteins and then extending along microtubules (sliding mechanism) or by attaching to the tips of growing microtubules [4]. New ER tubules can also be generated by hitchhiking on organelles that are transported along microtubules by molecular motors. Finally, recent work has shown that ER tubules can be pulled by shrinking microtubule ends [60]. The increased microtubule nucleation in the cells under ER stress shown in this work extends the microtubule network, which could help to expand ER membranes. However, new ER tubule branches can also appear without the involvement of microtubules in the process of de novo budding of the ER tubule from the existing ER tubule [60]. Therefore, one cannot rule out that microtubule-independent mechanisms of ER expansion are also activated in cells under ER stress.

C53 likely plays a role in the regulatory mechanism of microtubule nucleation in tunicamycin-treated cells since exogenous C53, expressed in C53_KO cells located to centrosomes from which it detached after tunicamycin treatment. Moreover, endogenous C53 relocated from the subcellular fraction containing centrosomes, and less than half of C53 remained associated with isolated centrosomes in tunicamycin treated cells relative to the control. Thus the relocation of C53 from the centrosome could unblock microtubule nucleation in cells under ER stress. Relocation of C53 is probably not induced by ufmylation of γ-tubulin, GCPs (GCP2, GCP4, GCP6), and C53. Ufmylation should change electrophoretic mobility of proteins, but we did not observe mobility changes for these proteins in cell lysates prepared in the presence of 5 mM N-ethylmaleimide, which blocks de-ufmylation activity, from tunicamycin-treated cells (unpublished results). UFL1-catalyzed ufmylation is, however, vital for relieving ER stress via ER-phagy [61]. An increase in centrosomal microtubule nucleation might facilitate increased autophagic flux, ER expansion, and relief of ER stress.

In conclusion, we show that UFL1 and C53 interacting with γTuRC proteins play an important role in microtubule nucleation. C53, whose protein level is modulated by UFL1, associates with the centrosome and represents a negative regulator of microtubule nucleation from the centrosomes. We demonstrate that the ER stress generated either by UFL1/C53 deletion or by pharmacological stressor stimulates microtubule nucleation via C53 displacement from the centrosome. The interaction of ER with newly formed microtubules could promote its enlargement to restore ER homeostasis. This suggests a novel mechanism for facilitating the ER network expansion under stress conditions.

## Figures and Tables

**Figure 1 cells-11-00555-f001:**
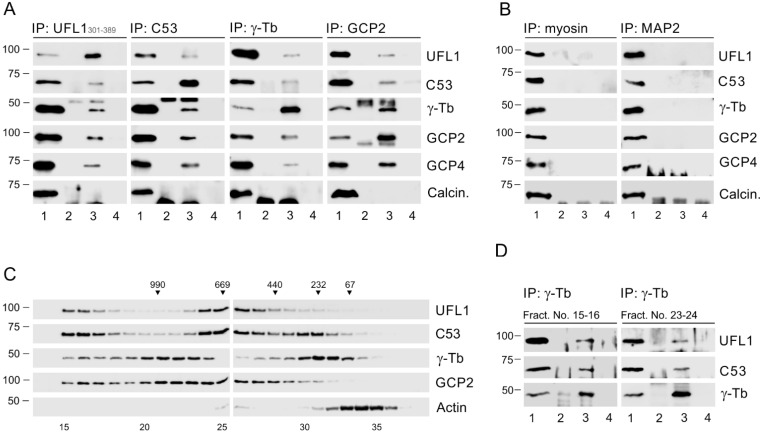
UFL1 and C53 interact with γTuRC proteins. (**A**) Immunoprecipitation experiments. Extracts from the membranous fraction (P2) of U2OS cells were precipitated with immobilized Abs specific to UFL1_301–389_, C53, γ-tubulin (γ-Tb), or GCP2. The blots were probed with Abs to UFL1, C53, γ-tubulin (γ-Tb), GCP2, GCP4, or calcineurin (Calcin.; negative control). Load (*lane 1*), immobilized Abs without cell extracts (*lane 2*), precipitated proteins (*lane 3*), and Ab-free carriers incubated with cell extracts (*lane 4*). (**B**) Isotype controls. Extracts from the membranous fraction (P2) of U2OS cells were precipitated with immobilized rabbit Ab to myosin or mouse mAb to MAP2 (IgG2b). Blots were probed with Abs to UFL1, C53, γ-tubulin (γ-Tb), GCP2, or GCP4. Load (*lane 1*), immobilized Abs not incubated with cell extracts (*lane 2*), precipitated proteins (*lane 3*), and carriers without Abs incubated with cell extracts (*lane 4*). (**C**) The size distribution of UFL1, C53, γ-tubulin (γ-Tb), GCP2, and actin in U2OS whole-cell extracts fractionated on the Superose 6 column. The calibration standards (in kDa) are indicated on the top. The numbers at the bottom denote individual fractions. (**D**) Pooled fractions (Nos. 15–16) and (Nos. 23–24) from fractionation shown in panel (**C**) were precipitated with Ab to γ-tubulin. The blots were probed with Abs to UFL1, C53, and γ-tubulin (γ-Tb). Load (*lane 1*), immobilized Abs without cell extracts (*lane 2*), precipitated proteins (*lane 3*), and Ab-free carriers incubated with cell extracts (*lane 4*).

**Figure 2 cells-11-00555-f002:**
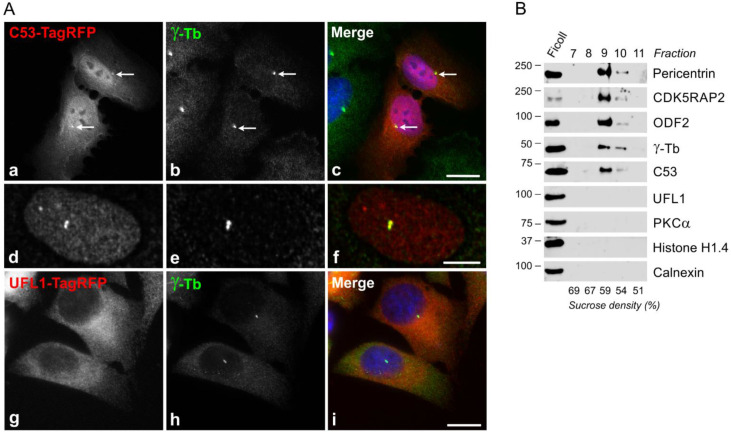
Subcellular localization of UFL1 and C53. (**A**) U2OS cells expressing TagRFP-tagged proteins were fixed and stained with Ab to γ-tubulin. Localization of C53-TagRFP (**a**,**d**) and γ-tubulin (**b**,**e**). Superposition of images (**c**,**f**) C53-TagRFP, red; γ-tubulin, green; DAPI, blue). Localization of UFL1-TagRFP (**g**) and γ-tubulin (**h**). Superposition of images (**i**, UFL1-TagRFP, red; γ-tubulin, green; DAPI, blue). Arrows indicate the same positions. Fixation D/F/M. Scale bar, 20 μm (**a**–**c**,**g**–**i**), 5 μm (**d**–**f**). (**B**) Association of proteins with purified centrosomes. Centrosomes enriched by centrifugation onto a Ficoll cushion were further purified by sucrose gradient centrifugation. The gradient was fractionated from the bottom. Individual fractions are indicated on the top, sucrose density in the fractions is shown at the bottom. Blots were probed with Abs to pericentrin, CDK5RAP2, ODF2, γ-tubulin (γ-Tb), C53, UFL1, PKCα, histone H1.4, and calnexin.

**Figure 3 cells-11-00555-f003:**
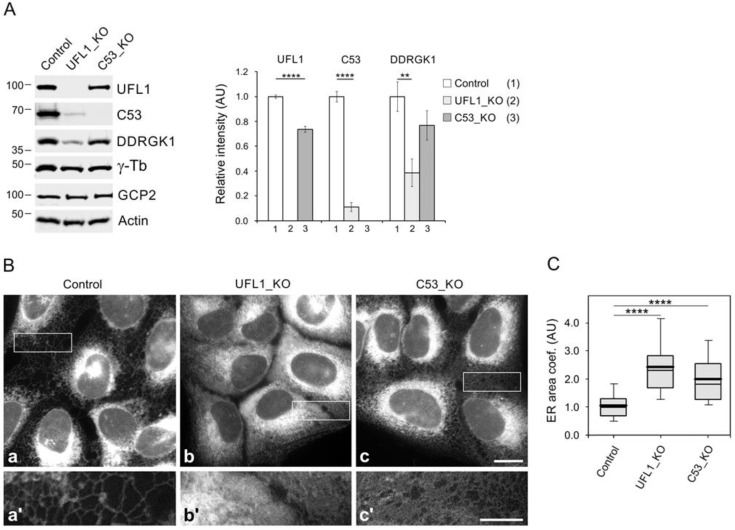
Characterization of cells lacking UFL1 or C53. (**A**) Changes in the expression of UFL1, C53, and DDRGK1. The blots from whole-cell lysates were probed with Abs to UFL1, C53, DDRGK1, γ-tubulin (γ-Tb), GCP2, and actin (loading control). Densitometric quantification of immunoblots is shown on the right. Relative intensities of corresponding proteins normalized to control cells and the amount of actin in individual samples. Values indicate mean ± SD (*n* = 3). One-way ANOVA with Sidak’s multiple comparisons test was performed to determine statistical significance. **, *p* < 0.01, ****, *p* < 0.0001. (**B**). Distribution of ER in control (**a**), UFL1_KO (**b**), and C53_KO (**c**) cells visualized by ER-Tracker in live cells. A higher magnification view of the boxed area is shown (**a’**–**c’**). The images (**a**–**c**) were collected and processed in the same manner. Scale bars, 20 µm (**c**) and 10 µm (**c’**). (**C**) ER area quantification in cells stained with ER-Tracker. ER area coefficients (area occupied by ER/area free of ER) were calculated from fluorescence intensities for ER-Tracker as described in the Section 2. The distributions of ER area coefficients (arbitrary units [AU]) are shown as box plots (three independent experiments, ≥ 15 cells counted for each experimental condition). Box plot of area coefficients in UFL1_KO (*n* = 85) and C53_KO cells (*n* = 89) relative to control cells (*n* = 93). The bottom and top of the box represent the 25th and 75th percentiles. Whiskers below and above the box indicate the 10th and 90th percentiles. One-way ANOVA with Dunnett’s multiple comparisons test was performed to determine statistical significance. ****, *p* < 0.0001.

**Figure 4 cells-11-00555-f004:**
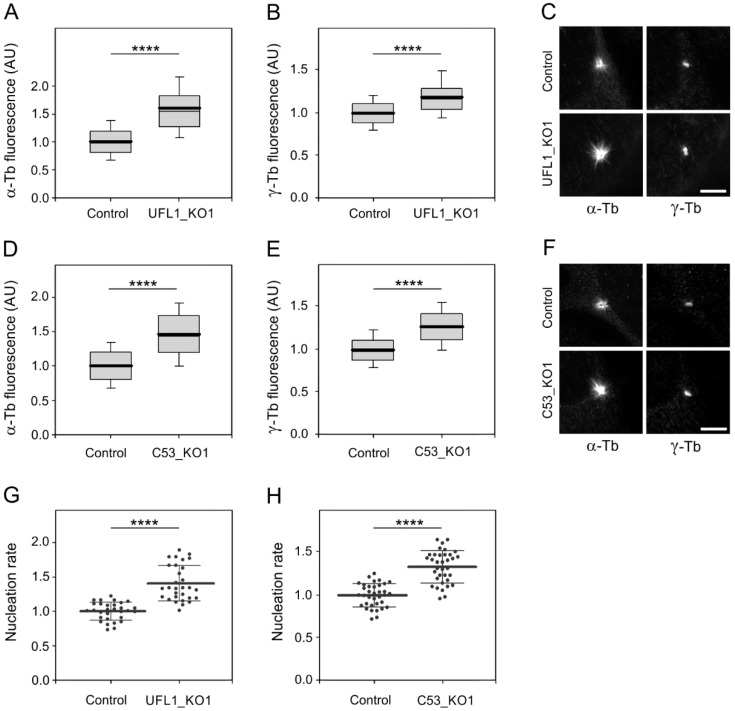
Deletion of UFL1 or C53 increases centrosomal microtubule nucleation. Centrosomal microtubule nucleation was evaluated by quantification of microtubule regrowth in fixed cells (**A**–**F**) and by measuring the microtubule nucleation rate in live cells (**G**,**H**). (**A**,**B**,**D**,**E**) The distribution of α-tubulin or γ-tubulin fluorescence intensities (arbitrary units [AU]) in 2-μm ROI at 2.0 min of microtubule regrowth are shown as box plots (three independent experiments, >58 cells counted for each experimental condition). (**A**,**B**) Box plot of α-tubulin (**A**) and γ-tubulin (**B**) fluorescence intensities in UFL1_KO1 cells (*n* = 239) relative to control cells (*n* = 237). (**D**,**E**) Box plot of α-tubulin (**D**) and γ-tubulin (**E**) fluorescence intensities in C53_KO1 cells (*n* = 257) relative to control cells (*n* = 274). The bold and thin lines within the box represent mean and median (the 50th percentile), respectively. The bottom and top of the box represent the 25th and 75th percentiles. Whiskers below and above the box indicate the 10th and 90th percentiles. (**C**,**F**) Labeling of α-tubulin and γ-tubulin in the microtubule regrowth experiment in the control and UFL1_KO cells (**C**) or the control and C53_KO1 cells (**F**). Cells were fixed (F/Tx/M) at 2.0 min of microtubule regrowth. The pairs of images (α-Tb), (γ-Tb) were collected and processed in the same manner. Scale bars, 5 μm. (**G**) Microtubule nucleation rate (EB3 comets/min) in UFL1_KO1 cells relative to controls. Three independent experiments (at least 10 cells counted in each experiment). Control (*n* = 30), UFL1_KO1 (*n* = 30). The bold and thin lines within the dot plot represent mean ± SD. (**H**) Microtubule nucleation rate (EB3 comets/min) in C53_KO1 cells relative to controls. Three independent experiments (at least 10 cells counted in each experiment). Control (*n* = 35), C53_KO1 (*n* = 35). The bold and thin lines within the dot plot represent mean ± SD. Two-tailed, unpaired Student’s *t*-test was performed to determine statistical significance. ****, *p* ˂ 0.0001.

**Figure 5 cells-11-00555-f005:**
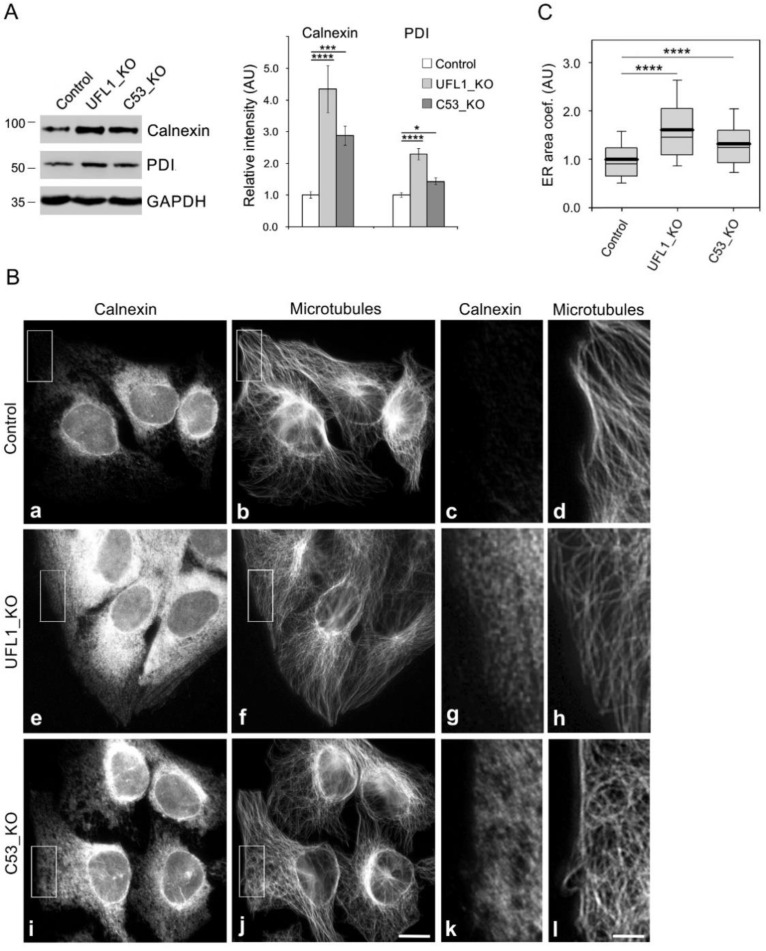
Deletion of UFL1 or C53 induces UPR and expansion of ER. (**A**) Immunoblot analysis of calnexin and PDI in whole-cell lysates of U2OS cells lacking UFL1 or C53. GAPDH served as a loading control. Densitometric quantification of immunoblots is shown on the right. Relative intensities of corresponding proteins normalized to control cells and the amount of GAPDH in individual samples. Values indicate mean ± SD (*n* = 4 for calnexin; *n* = 3 for PDI). (**B**) Immunofluorescence microscopy. (**a**–**d**) Control U2OS cells, (**e**–**h**) UFL1-deficient cells (UFL1_KO) and (**i**–**l**) C53-deficient cells (C53_KO). Cells were fixed and double-labeled for calnexin (**a**,**e**,**i**) and β-tubulin (**b**,**f**,**j**; Microtubules). Higher magnification views of the regions delimited by rectangles are shown on the right of images from control (**c**,**d**), UFL1_KO (**g**,**h**), and C53_KO (**k**,**l**) cells. The images (**a**,**e**,**i**) and (**c**,**g**,**k**) were collected and processed in the same manner. Fixation F/Tx. Scale bars, 20 μm (**j**), and 5 µm (**l**). (**C**) ER area quantification in fixed cells stained with Ab to calnexin. ER area coefficients (area occupied by ER/area free of ER) were calculated from fluorescence intensities for calnexin. The distributions of ER area coefficients (arbitrary units [AU]) are shown as box plots (four independent experiments, ≥16 cells counted for each experimental condition). Box plot of area coefficients in UFL1_KO (*n* = 111) and C53_KO cells (*n* = 127) relative to control cells (*n* = 149). The bottom and top of the box represent the 25th and 75th percentiles. Whiskers below and above the box indicate the 10th and 90th percentiles. (**A**,**C**) One-way ANOVA with Dunnett’s multiple comparisons test was performed to determine statistical significance. *, *p* < 0.05, ***, *p* < 0.001, ****, *p* < 0.0001.

**Figure 6 cells-11-00555-f006:**
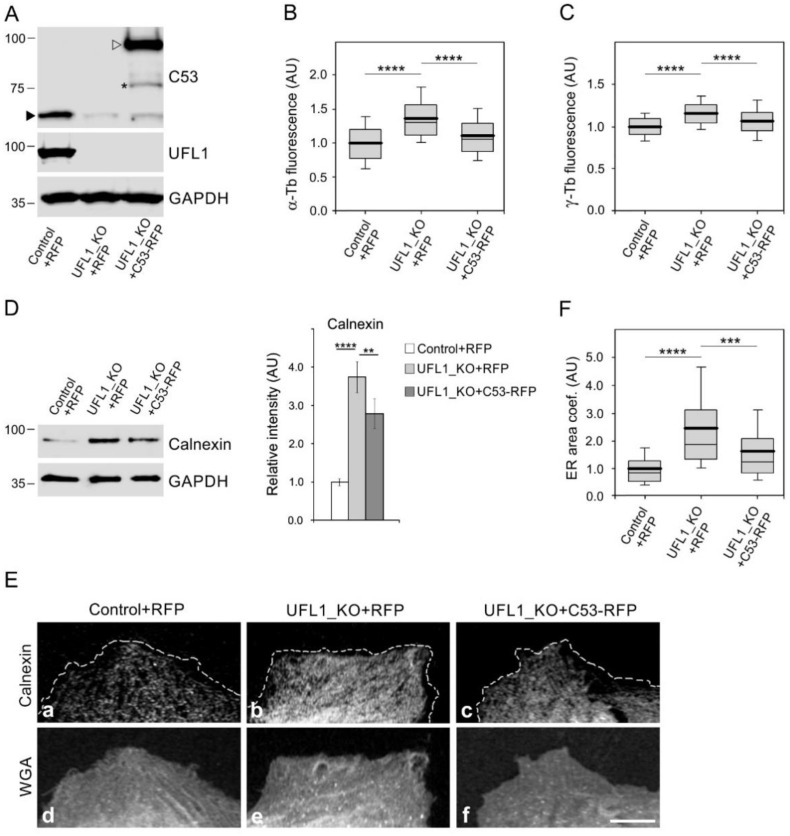
C53 is sufficient to attenuate centrosomal microtubule nucleation and ER expansion in *UFL1* knockout cells. Phenotypic rescue experiment with control cells expressing TagRFP (Control+RFP), UFL1_KO cells expressing TagRFP (UFL1_KO+RFP), and UFL1_KO cells rescued by C53-TagRFP (UFL1_KO+C53-RFP). (**A**) Immunoblot analysis of C53 and UFL1 in whole-cell lysates. GAPDH served as the loading control. Black and empty arrowheads and asterisk denote, respectively, endogenous C53, C53-TagRFP, and its fragment. (**B**,**C**) The distributions of α-tubulin or γ-tubulin fluorescence intensities (arbitrary units [AU]) in 2-μm ROI at 2.0 min of microtubule regrowth are shown as box plots (three independent experiments, >37 cells counted for each experimental condition). Box plot of α-tubulin (**B**) and γ-tubulin (**C**) fluorescence intensities in UFL1_KO+RFP (*n* = 133) and UFL1_KO+C53-RFP (*n* = 152) relative to Control+RFP (*n* = 174). (**D**) Immunoblot analysis of calnexin in whole-cell lysates. GAPDH served as the loading control. Densitometric quantification of immunoblots is shown on the right. Relative intensities of corresponding proteins normalized to control cells and the amount of GAPDH in individual samples. Values indicate mean ± SD (*n* = 4). (**E**) Immunofluorescence microscopy of fixed cells stained with Ab to calnexin to mark ER (**a**–**c**) and AlexaFluor 647-conjugated WGA to delineate cell boundary (**d**–**f**). The images (**a**–**c**) were collected and processed in the same manner. Fixation F/Tx. Scale bar, 10 μm. (**F**) ER area quantification. The distributions of ER area coefficients (area occupied by ER/area free of ER; arbitrary units [AU]) are shown as box plots (three independent experiments, ≥20 cells counted for each experimental condition). Box plot of area coefficients in UFL1_KO+RFP (*n* = 91) and UFL1_KO+C53-RFP (*n* = 94) relative to Control+RFP (*n* = 147). (**B**,**C**,**F**) Bold and thin lines within the box represent mean and median (the 50th percentile), respectively. The bottom and top of the box represent the 25th and 75th percentiles. Whiskers below and above the box indicate the 10th and 90th percentiles. (**B**–**D**,**F**) One-way ANOVA with Sidak’s multiple comparisons test was performed to determine statistical significance. **, *p* < 0.01, ***, *p* < 0.001, ****, *p* < 0.0001.

**Figure 7 cells-11-00555-f007:**
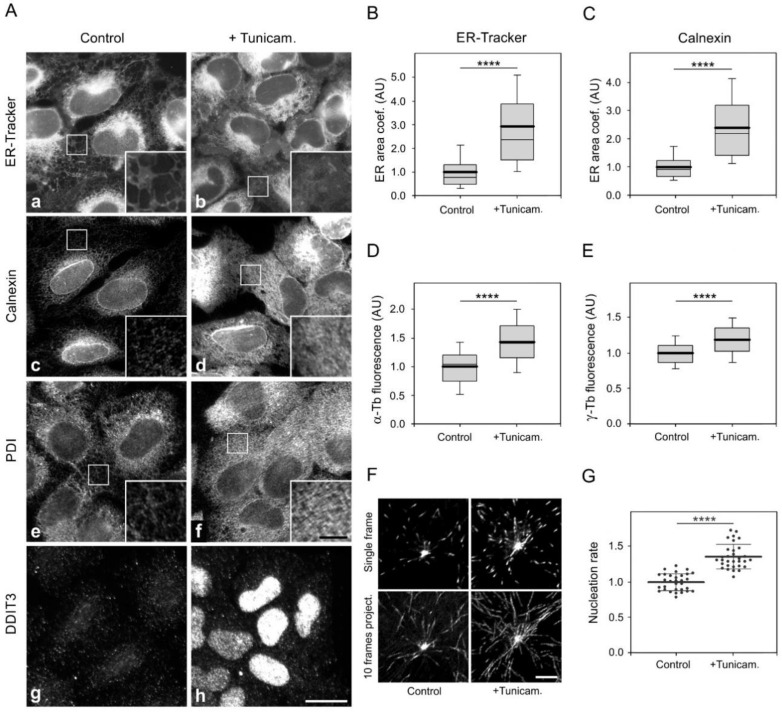
Generation of ER stress by tunicamycin increases centrosomal microtubule nucleation. U2OS cells were treated with 1 µg/mL tunicamycin (+Tunicam.) or DMSO carrier (Control) for 24 h. (**A**). Effect of tunicamycin on ER expansion and subcellular distribution of calnexin, PDI, and ER stress-induced transcription factor DDIT3. (**a**,**b**) Visualization of ER in live cells by ER-Tracker. (**c**–**h**) Fixed cells stained for calnexin (**c**,**d**), PDI (**e**,**f**), and DDIT3 (**g**,**h**). Insets represent an enlargement of the boxed area. Fixation F/Tx. Pairs of images (**a**,**b**), (**c**,**d**), (**e**,**f**), and (**g**,**h**) were collected and processed in the same manner. Scale bars, 20 μm (**h**) and 5 µm (inset in **f**). (**B**) ER area quantification in live cells stained with ER-Tracker. The distributions of ER area coefficients (area occupied by ER/area free of ER; arbitrary units [AU]) are shown as box plots (three independent experiments, ≥23 cells counted for each experimental condition). Box plot of ER area coefficients in tunicamycin-treated cells (*n* = 81) relative to control cells (*n* = 108). (**C**) ER area quantification in fixed cells stained with Ab to calnexin. The distributions of ER area coefficients (area occupied by ER/area free of ER; arbitrary units [AU]) are shown as box plots (three independent experiments, ≥19 cells counted for each experimental condition). Box plot of ER area coefficients in tunicamycin-treated cells (*n* = 80) relative to control cells (*n* = 65). (**D**,**E**) The distributions of α-tubulin or γ-tubulin fluorescence intensities (arbitrary units [AU]) in 2-μm ROIs at 3.0 min of microtubule regrowth in control and tunicamycin-treated cells are shown as box plots (four independent experiments, >27 cells counted for each experimental condition). Box plot of α-tubulin (**D**) and γ-tubulin (**E**) fluorescence intensities in tunicamycin-treated cells (*n* = 234) relative to control cells (*n* = 181). (**F**) Time-lapse imaging of control and tunicamycin-treated cells expressing EB3-mNeonGreen. Still images of EB3 (Single frame) and tracks of EB3 comets over 10 s (10 frames project.). Scale bar, 5 µm. (**G**) Microtubule nucleation rate (EB3 comets/min) in tunicamycin-treated cells relative to control cells. Three independent experiments (at least 9 cells counted in each experiment). Control (*n* = 31), tunicamycin-treated cells (*n* = 31). The bold and thin lines within the dot plot represent mean ± SD. (**B**–**E**) The bold and thin lines within the box represent mean and median (the 50th percentile), respectively. The bottom and top of the box represent the 25th and 75th percentiles. Whiskers below and above the box indicate the 10th and 90th percentiles. (**B**–**E**,**G**) Two-tailed, unpaired Student’s *t*-test was performed to determine statistical significance. **** *p* < 0.0001.

**Figure 8 cells-11-00555-f008:**
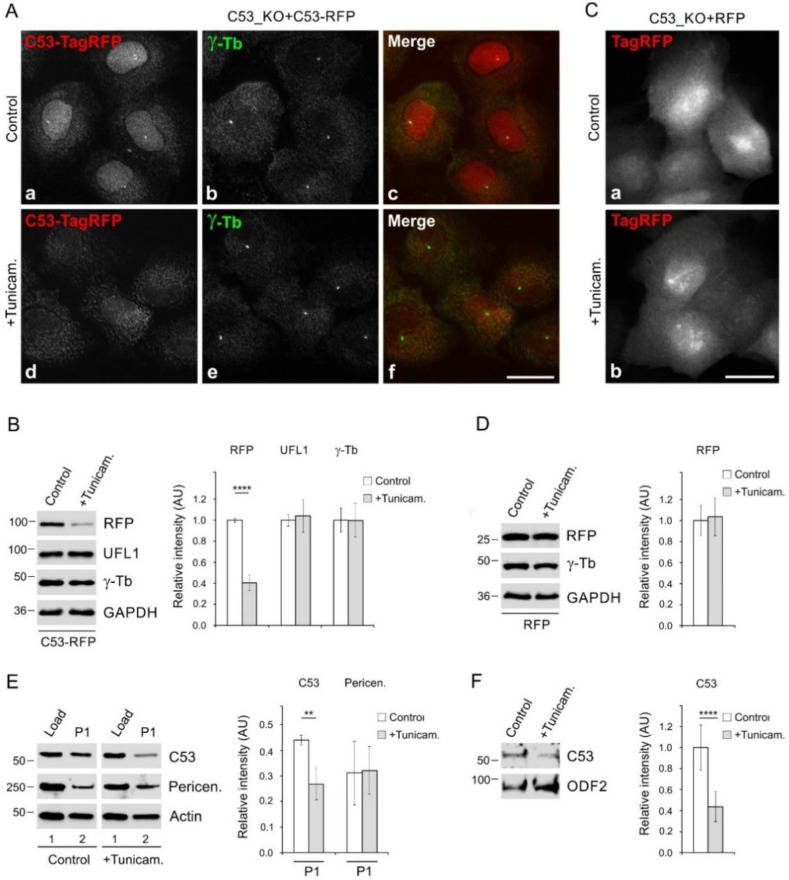
Tunicamycin affects the distribution of C53. Cells were treated with 1 µg/mL tunicamycin (+Tunicam.) or DMSO carrier (Control) for 24 h. (**A**,**B**) C53_KO cells expressing C53-TagRFP (C53_KO+C53-RFP). (**A**) Immunofluorescence microscopy of fixed cells stained for γ-tubulin. (**a**–**c**) Control cells. C53-TagRFP (**a**), γ-tubulin (**b**), superposition of images (**c**, C53-TagRFP, red; γ-tubulin, green). (**d**–**f**) Tunicamycin-treated cells. C53-TagRFP (**d**), γ-tubulin (**e**), superposition of images (**f**, C53-TagRFP, red; γ-tubulin, green). Images (**a**,**d**) and (**b**,**e**) were collected and processed in the exact same manner. Fixation Tx/F/M. Scale bar, 20 µm. (**B**) Immunoblot analysis of whole-cell lysates with Abs to RFP, UFL1, γ-tubulin (γ-Tb), and GAPDH (loading control). Densitometric quantification of immunoblots is shown on the right. Relative intensities of corresponding proteins normalized to control cells and the amount of GAPDH in individual samples. Values indicate mean ± SD (*n* = 4). (**C**,**D**) C53-KO cells expressing TagRFP (C53_KO+RFP). (**C**) Immunofluorescence microscopy of fixed control cells (**a**) and tunicamycin-treated cells (**b**). Images (**a**,**b**) were collected and processed in exactly the same manner. Fixation F/Tx. Scale bar, 20 µm. (**D**) Immunoblot analysis of whole-cell lysates with Abs to RFP, γ-tubulin (γ-Tb), and GAPDH (loading control). Densitometric quantification of immunoblots is shown on the right. Relative intensities of corresponding proteins normalized to control cells and the amount of GAPDH in individual samples. Values indicate mean ± SD (*n* = 3). (**E**) Distribution of proteins in fractions after differential centrifugation of the cell homogenate. Cell fractions were prepared as described in the Section 2. Cell homogenate (*lane 1*), pellet P1 (*lane 2*). Immunoblot analysis with Abs to C53, pericentrin, and actin (loading control). Densitometric quantification of immunoblots is shown on the right. Intensities of corresponding proteins in P1 normalized to loads (relative intensity 1.0). Values indicate mean ± SD (*n* = 4). (**F**) Association of proteins with purified centrosomes. Immunoblot analysis with Abs to C53 and centrosomal protein ODF2 (loading control). Densitometric quantification of immunoblots is shown on the right. Relative intensity of C53 normalized to control cells and the amount of ODF2. Values indicate mean ± SD (*n* = 7). A two-tailed, unpaired Student’s *t*-test was performed to determine statistical significance. **, *p* < 0.01; **** *p* < 0.0001.

## Data Availability

The datasets generated during and/or analyzed during the current study are available from the corresponding author on request. The original images for blots and gels presented in this study are in the Appendix A here.

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
