# Peer review of "C53 Interacting with UFM1-Protein Ligase 1 Regulates Microtubule Nucleation in Response to ER Stress"

_cells, 2022, doi:10.3390/cells11030555_

Round 1

Reviewer 1 Report

In this manuscript, Klebanovych et al., describe the role of C53 and UFL1 in controlling microtubule nucleation through association with g-tubulin at the centrosomes.

Single knockout of UFL1 leads to decreased levels of C53, expansion of ER compartment and increased microtubule nucleation. Similar phenotype is observed upon single knockout of C53. Finally, pharmacological induction of ER stress via Tunicamycin treatment decrease the levels of C53 and causes an expansion of the ER compartment accompanied by enhanced microtubule nucleation. The data presented are interesting and add further knowledge on the already described connection between C53, UFL1 and ER stress. However, some concerns need to be address in order to corroborate and clarify the presented mechanism.

Major points

In figure 5 the authors claim that genetic loss of UFL1 and C53 induces UPR. However, in order to prove this, expression of proteins of the UPR branches (phosphoPERK/CHOP, IRE1/sXBP1 or ATF6) should be measured via western blot. Infact, the increase of Calnexin and PDI shown in Figure 5A might be due to ER expansion, but not necessarily to ER stress.

In figure 8 the authors show that cells undergoing ER stress show increased microtubule nucleation that facilitates ER expansion, and this partly occur through declined levels of C53.  Is ULF1 involved in this mechanism? To confirm the mechanism described in the paper, the authors should perform Tunicamycin treatment in WT cells or cells overexpressing ULF1 and analyze microtubule nucleation and C53 levels.

As general information, what is phenotype of U2OS cells knockout for UFL1 and C53? Do they show defect in proliferation and cell cycle? This info should be added in the supplementary information.

Minor points

Microscopy images showing ER tracker (Fig. 3b) should be quantified.

References in line 920, 939, 998 should be added to the reference list.

Author Response

REVIEWER #1

In this manuscript, Klebanovych et al., describe the role of C53 and UFL1 in controlling microtubule nucleation through association with g-tubulin at the centrosomes.

Single knockout of UFL1 leads to decreased levels of C53, expansion of ER compartment and increased microtubule nucleation. Similar phenotype is observed upon single knockout of C53. Finally, pharmacological induction of ER stress via Tunicamycin treatment decrease the levels of C53 and causes an expansion of the ER compartment accompanied by enhanced microtubule nucleation. The data presented are interesting and add further knowledge on the already described connection between C53, UFL1 and ER stress. However, some concerns need to be address in order to corroborate and clarify the presented mechanism.

We thank the reviewer for his thoughts and comments and have followed his recommendations.

Major points

1/

In figure 5 the authors claim that genetic loss of UFL1 and C53 induces UPR. However, in order to prove this, expression of proteins of the UPR branches (phosphoPERK/CHOP, IRE1/sXBP1 or ATF6) should be measured via western blot. In fact, the increase of Calnexin and PDI shown in Figure 5A might be due to ER expansion, but not necessarily to ER stress.

 Response:

To prove that genetic loss of UFL1 and C53 activates the UPR, we performed qRT-PCR analysis of genes involved in the UPR. Deletion of UFL1 resulted in significant upregulation of the ER stress sensors IRE1α (ERN1) and ATF6α (ATFA) and the downstream target Grp78/BIP (HSPA5) and the downstream target CHOP (DDIT3) of the PERK stress sensor. Deletion of C53 resulted in upregulation of IRE1α and Grp78/BIP (Figure S5D in the revised manuscript). In addition, XBP1 mRNA splicing assay showed that the spliced variant XBP1 (XBP1s) involved in IRE1α signaling was clearly detectable in both UFL1_KO and C53_KO cells (Figure S5E in the revised manuscript). These results document the UPR in prepared cell lines and suggest that the PERK branch may not be activated in cells lacking C53, as has been previously reported for intestinal cells {Quintero et al, Cell Death and Disease 12:131, 2021).

We have changed the text in Materials and Methods accordingly (2.8. Real-time qRT-PCR, page 6; 2.9. XBP1 mRNA splicing assay, page 6) and added a new table summarising the primers used for qRT-PCR (Table S1 in the revised manuscript). The new results are described in the Results section (page 13, lines 573-581).

2/

In figure 8 the authors show that cells undergoing ER stress show increased microtubule nucleation that facilitates ER expansion, and this partly occur through declined levels of C53.  Is ULF1 involved in this mechanism? To confirm the mechanism described in the paper, the authors should perform Tunicamycin treatment in WT cells or cells overexpressing ULF1 and analyze microtubule nucleation and C53 levels.

Response:

In the original manuscript submission, we proposed that relocation of endogenous C53 from the centrosome in tunicamycin-treated cells results in increased microtubule nucleation. To determine whether UFL1 has an effect on microtubule nucleation and C53 levels in tunicamycin-treated cells, we performed additional experiments and first compared nucleation in cells overexpressing UFL1-TagRFP or TagRFP alone (control). Using time-lapse imaging, we found comparable nucleation rates (Figure S8D in the revised manuscript). Endogenous C53 levels were not affected by overexpression of UFL1-TagRFP (Figure S3A) or tunicamycin treatment (Figure S8E in the revised manuscript). Overexpression of UFL1-TagRFP also did not alter microtubule nucleation in tunicamycin-treated cells (Figure S8 in the revised manuscript). These results suggest that UFL1 is not involved in the mechanism regulating microtubule nucleation in tunicamycin-treated cells.

The new results are described in the Results section (page 20, lines 796-801).

3/

 As general information, what is phenotype of U2OS cells knockout for UFL1 and C53? Do they show defect in proliferation and cell cycle? This info should be added in the supplementary information.

Response:

To assess the effect of deletion of UFL1 or C53 on cell division, we determined cell growth in control and UFL1_KO or C53_KO cells. Compared with control cells, the number of viable cells decreased significantly in both UFL1-KO and C53_KO, but proliferation was more impaired in UFL1_KO cells (Figure S5A in the revised manuscript). Analysis of asynchronous cell cultures by FACS analysis revealed a trend toward more cells in G1 phase and fewer in G2/M phase for UFL1_KO and C53_KO cells (Figure S5B in the revised manuscript).

We have changed the text in Materials and Methods accordingly (2.15. Evaluation of cell growth and FACS analysis, page 8). The new results are described in the Results section (page 12, lines 560-564).

Minor points

1/

Microscopy images showing ER tracker (Fig. 3b) should be quantified.

We quantified the distribution of ER tracker in control, UFL1_KO, and C53_KO cells (Figure 3A in the revised manuscript).

The new results are described in the Results section (page 13, lines 587-590).

2/

References in line 920, 939, 998 should be added to the reference list.

We added references to the reference list.

3/

English language and style are fine/minor spell check required.

We spell-checked our text.

Reviewer 2 Report

The manuscript " C53 interacting with UFM1-protein ligase 1 regulates microtubule nucleation in response to ER stress “by Klebanovych et al. reports that C53 and UFL1 play a role in microtubule nucleation. Initially, the authors showed, using IP experiments and mass spec analysis, that UFL1 is an interactor of gamma-tubulin.  Then, they generated KO cells of UFL1 and C53 and tested the effect on ER distribution. Specifically, while in the WT cells the tubules at the cell periphery were sparse, in UFL1_KO cells, prominent formation of ER tubules was observed.  In addition, they found that deletion of UFL1 or C53 increases centrosomal microtubule nucleation

This work advances our understanding of centrosomal microtubule nucleation. However, the data supporting the conclusions are not always convincing and, in some cases, require further validation.

It was previously shown that C53 interacts with UFL1 as well as with gamma tubulin. Therefore, the novelty of showing that UFL1 is in complex with gamma tubulin is not clear. Can the authors test using pure proteins whether UFL1 directly interacts with gamma tubulin or whether this interaction is mediated by C53? Alternatively, in the gel filtration experiment, it will be nice to show the elution profile of UFL1 in cells that lack C53.  

Another question is regarding UFBP1.  Does regulation of microtubule nucleation by UFL1 require UFBP1? Specifically, is the effect of UFBP1 KO similar to that of UFL1 KO? 

Author Response

REVIEWER #2

The manuscript " C53 interacting with UFM1-protein ligase 1 regulates microtubule nucleation in response to ER stress “by Klebanovych et al. reports that C53 and UFL1 play a role in microtubule nucleation. Initially, the authors showed, using IP experiments and mass spec analysis, that UFL1 is an interactor of gamma-tubulin.  Then, they generated KO cells of UFL1 and C53 and tested the effect on ER distribution. Specifically, while in the WT cells the tubules at the cell periphery were sparse, in UFL1_KO cells, prominent formation of ER tubules was observed.  In addition, they found that deletion of UFL1 or C53 increases centrosomal microtubule nucleation. This work advances our understanding of centrosomal microtubule nucleation. However, the data supporting the conclusions are not always convincing and, in some cases, require further validation.

We thank the reviewer for his/her deep insight and pertinent comments and have followed his/her recommendations.

1/

It was previously shown that C53 interacts with UFL1 as well as with gamma tubulin. Therefore, the novelty of showing that UFL1 is in complex with gamma tubulin is not clear.

We agree that the direct interaction between UFL1 and C53 is well established. It has been previously reported that C53 forms complexes with nuclear γ-tubulin (HoÅ™ejší et al., J. Cell Physiol. 227:367, 2012), but it was unclear whether C53 can associate with γ-tubulin in the other cell parts and whether its complexes contain GCPs essential for γTuRC-dependent microtubule nucleation. The results presented suggest that multiprotein complexes containing UFL1/C53 and γTuRC proteins occur at different cellular locations in different cell types. We emphasise this point more clearly in Discussion (page 20, lines 829-832).

2/

Can the authors test using pure proteins whether UFL1 directly interacts with gamma tubulin or whether this interaction is mediated by C53?

To determine whether UFL1 and C53 interact directly with γ-tubulin, we performed pull-down assays with GST-γ-tubulin and purified FLAG-tagged UFL1, C53, or nucleophosmin (NPM1, negative control). C53 clearly bound to γ-tubulin, in contrast to UFL1. Control NPM1 did not bind to GST-γ-tubulin (Figure S2G in the revised manuscript). These data suggest that C53 may be able to bind γ-tubulin directly and mediate binding of UFL1.

We have changed the Materials and Methods accordingly (page 3, lines 97-98; page 4, lines 164-168; page 7, lines 335-337). The new results are described in the Results section (pages 10-11; lines 498-518).

3/

Alternatively, in the gel filtration experiment, it will be nice to show the elution profile of UFL1 in cells that lack C53.

The size distribution of UFL1 in control and C53_KO whole-cell extracts was comparable, suggesting that the formation of large UFL1 complexes is not exclusively dependent on the presence of C53 (Figure S5C in the revised manuscript).

We have changed the Materials and Methods accordingly (page 7, line 318). The new results are described in the Results section (pages 13; lines 569-571).

4/

Another question is regarding UFBP1.  Does regulation of microtubule nucleation by UFL1 require UFBP1? Specifically, is the effect of UFBP1 KO similar to that of UFL1 KO? 

To determine whether DDRGK1 (UFBP1) plays a role in microtubule nucleation, we efficiently depleted DDRGK1 by two siRNAs in U2OS cells expressing mNeonGreen-tagged EB3 (Figure S6H in the revised manuscript). Using time-lapse imaging, we detected no difference in nucleation rates in control and DDRGK1-depleted cells (Figure S6I in the revised manuscript). This indicates that DDRGK1 does not participate in the regulation of centrosomal microtubule nucleation.

 We have correspondingly changed the text in Material and methods (2.7. RNA interference, pages 27-28). The new results are described in the Results section (pages 14-15, lines 640-663).

Round 2

Reviewer 1 Report

The authors addressed my comments and therefore I agree to publish the paper in the current format.

Reviewer 2 Report

I fully support the publication of this work.  The authors addressed my concerns